# Projection-Free Online Convex Optimization via Efficient Newton Iterations

**Khashayar Gatmiry**
MIT
gatmiry@mit.com

**Zakaria Mhammedi**
MIT
mhammedi@mit.edu

## Abstract

This paper presents new projection-free algorithms for Online Convex Optimization (OCO) over a convex domain $\mathcal{K} \subset \mathbb{R}^d$. Classical OCO algorithms (such as Online Gradient Descent) typically need to perform Euclidean projections onto the convex set $\mathcal{K}$ to ensure feasibility of their iterates. Alternative algorithms, such as those based on the Frank-Wolfe method, swap potentially-expensive Euclidean projections onto $\mathcal{K}$ for linear optimization over $\mathcal{K}$. However, such algorithms have a sub-optimal regret in OCO compared to projection-based algorithms. In this paper, we look at a third type of algorithms that output approximate Newton iterates using a self-concordant barrier for the set of interest. The use of a self-concordant barrier automatically ensures feasibility without the need of projections. However, the computation of the Newton iterates requires a matrix inverse, which can still be expensive. As our main contribution, we show how the stability of the Newton iterates can be leveraged to only compute the inverse Hessian a vanishing fractions of the rounds, leading to a new efficient projection-free OCO algorithm with a state-of-the-art regret bound.

## 1 Introduction

We consider the Online Convex Optimization (OCO) problem over a convex set $\mathcal{K} \subset \mathbb{R}^d$, in which a learner (algorithm) plays a game against an adaptive adversary for $T$ rounds. At each round $t \in [T]$, the learner picks $w_t \in \mathcal{K}$ given knowledge of the history $\mathcal{H}_{t-1} \coloneqq \{(\ell_s, w_s)\}_{s<t}$. Then, the adversary picks a convex loss function $\ell_t : \mathcal{K} \to \mathbb{R}$ with the knowledge of $\mathcal{H}_{t-1}$ and the iterate $w_t$, and the learner suffers loss $\ell_t(w_t)$ and proceeds to the next round. The goal of the learner is to minimize the regret after $T$ rounds:

$$\mathrm{Reg}_T(w) = \sum_{t=1}^{T} \ell_t(w_t) - \sum_{t=1}^{T} \ell_t(w),$$

against any comparator $w \in \mathcal{K}$. The aim of this paper is to design computationally-efficient (projection-free) algorithms for OCO that enjoy the optimal (up to log-factor in $T$) $\widetilde{O}(\sqrt{T})$ regret.

The OCO framework captures many optimization settings relevant to machine learning applications. For example, OCO algorithms can be used in offline convex optimization as more computationally-and memory-efficient alternatives to interior-point and cutting plane methods whenever the dimension $d$ is large [14, 16]. OCO algorithms are also often used in stochastic convex optimization, where the standard $O(\sqrt{T})$ regret (achieved by, e.g. Online Gradient Descent) translates into the optimal $O(1/\sqrt{T})$ rate[1] via the classical online-to-batch conversion technique [3, 37]. It has been shown that OCO algorithms can also achieve state-of-the-art accelerated rates in both the offline and stochastic optimization settings despite being designed for the more general OCO framework [6, 28]. What is

---

[1]This is the optimal rate when no further assumptions are made.

37th Conference on Neural Information Processing Systems (NeurIPS 2023).

more, it has recently been shown that even non-convex (stochastic) optimization can be reduced to online linear optimization (a special case of OCO), where it is then possible to recover the best-known convergence rates for the setting [7].

Given the prevalent use of OCO algorithms in machine learning applications, it is important to have computationally-efficient algorithms that scale well with the dimension $d$ of the ambient space. However, most OCO algorithms fall short of being efficient because of the need of performing (Euclidean) projections onto $\mathcal{K}$ (potentially at each iteration) to ensure that the iterates are feasible. These projections are often inefficient, especially in high-dimensional settings with complex feasible sets. Existing projection-free OCO algorithms address this computational challenge by swapping potentially-expensive Euclidean projections for often much cheaper linear optimization or separation over the feasible set $\mathcal{K}$. However, existing projection-free algorithms have sub-optimal regret guarantees in terms of their dependence in $T$, or have potentially unbounded "condition numbers" for the feasible set multiplying their regret guarantee.

**Contributions.** In this paper, we address these computational and performance challenges by revisiting an existing (but somewhat overlooked) type of projection-free OCO algorithms. Unlike existing algorithms, our proposed method does not require linear optimization or separation over the feasible set $\mathcal{K}$. Instead, the algorithm, Barrier-Regularized Online Newton Step (`BARONS`),[2] uses a self-concordant barrier $\Phi$ for the set $\mathcal{K}$ to always output iterates that are guaranteed to be within $\mathcal{K}$; much like interior point methods for offline optimization. In particular, our algorithm outputs Newton iterates with respect to time-varying, translated versions of $\Phi$. The main novelty of our work is in devising a new efficient way of computing the Newton iterates without having to evaluate the inverse of the Hessian of the barrier at every iteration, which can be computationally expensive in high-dimensional settings. Our algorithm only needs to compute a full inverse of the Hessian a vanishing $O(1/\sqrt{T})$ fraction of the rounds. For the rest of the rounds, the computational cost is dominated by that of evaluating the gradient of the barrier $\Phi$, which can be much cheaper than evaluating the inverse of its Hessian in many cases.

For the special case of a polytope with $m$ constraints, we show that there is a choice of a barrier (e.g. the Lee-Sidford barrier) that when used within our algorithm, reduces the per-round computational cost to essentially $\widetilde{O}(1)$ linear-system-solves of size $m \times d$. We show that this is often cheaper than performing linear optimization over $\mathcal{K}$, which other projection-free algorithms require. More importantly, our algorithm achieves a *dimension-free* $\widetilde{O}(\sqrt{T})$ regret bound. This improves over the existing regret bounds of projection-free algorithms over polytopes. For example, among projection-free algorithms that achieve a $O(\sqrt{T})$ regret, the algorithms by [28, 11, 26], which require a separation/membership Oracle for $\mathcal{K}$, have a multiplicative $\kappa = R/r$ factor multiplying their regret bounds, where $r, R > 0$ are such that $\mathcal{B}(r) \subseteq \mathcal{K} \subseteq \mathcal{B}(R)$. The constant $\kappa$, known as the *asphericity* [12], can in principle be arbitrarily large. Even after applying a potentially expensive pre-processing step, which would typically involve putting the set $\mathcal{K}$ into (near-) isotropic position [9, 38], $\kappa$ can still be as large as $d$ in the worst-case, and so the regret bounds achieved by the algorithms of [28, 11, 26] can be of order $O(d\sqrt{T})$; this is worse than ours by a $d$ factor. Other projection-free algorithms based on the Frank-Wolfe method, e.g. those in [10, 35, 19], also have multiplicative condition numbers that are even less benign that the asphercity $\kappa$. In fact, the condition numbers in the regret bounds for polytopes appearing in, e.g. [10], can in principle be arbitrarily large regardless of any pre-processing.

Finally, another advantage of our algorithm is that it can guarantee a sublinear regret even for non-Lipschitz losses (i.e. where the norm of the sub-gradients may be unbounded). In particular, we show that the general guarantee of `BARONS` implies a $\widetilde{O}(\sqrt{dT})$ regret bound for the portfolio selection problem [5] and a problem of linear prediction with log-loss [36], all while keeping the per-round computational cost under $\widetilde{O}(d^2)$, when $T \geq d$. The losses in both of these problems are neither bounded or Lispchitz.

**Related works.** In the past decade, many projection-free OCO algorithms have been developed to address the computational shortcoming of their projection-based counter parts [14, 16, 15, 18, 28, 11]. Most projection-free algorithms are based on the Frank-Wolfe method and perform linear optimization (typically once per round) over $\mathcal{K}$ instead of Euclidean projection. Under no additional assumptions

---

[2]We credit the name `BARONS` to [27] who used barrier-regularized Newton steps for the portfolio selection problem.

other than convexity and lipschitzness of the losses, the best-known regret bound for such algorithms scales as $O(T^{3/4})$ [15]. While this bound is still sublinear in $T$ and has no dependence in the dimension $d$, it is sub-optimal compared to the $O(\sqrt{T})$ regret bound achievable with projection-based algorithms. In the recent years, there have been improvements to this bound under additional assumptions such as when the functions are smooth and/or strongly convex [15, 18], or when the convex set $\mathcal{K}$ is smooth and/or strongly convex [1, 23, 29, 24]. For the case where $\mathcal{K}$ is a polytope, [10] presented a linear-optimization-based algorithm that enjoys a $O(\mu\sqrt{dT})$ regret bound, where $\mu$ is a conditioning number for the set $\mathcal{K}$. Unfortunately, $\mu$ can be large for many sets of interests as it essentially scales inversely with the minimum distance between the vertices of $\mathcal{K}$. In this work, we achieve a *dimension-free* $\widetilde{O}(\sqrt{T})$ regret bound without the $\mu$ factor.

More recently a new type of projection-free algorithms have emerged which use membership/separation oracle calls instead of linear optimization [28, 11, 24, 26]. From a computational perspective, separation-based and linear optimization-based algorithms are not really comparable, since there are sets over which separation is cheaper than linear optimization, and vice-versa. On the regret side, separation-based algorithms have been show to achieve a $O(\kappa\sqrt{T})$ regret bound, where $\kappa$ is the asphercity of the set $\mathcal{K}$. Separation-based algorithms are simple, often easy to analyze, and achieve the optimal-in-$T$ regret bound, unlike linear optimization-based algorithms. However, the multiplicative factor $\kappa$ in their regret bounds means that a pre-conditioning step may be required to ensure it is appropriately bounded. This precondition step would involve putting the set into (near-) isotropic position [9]; an operation, that can cost $\widetilde{O}(d^4)$ arithmetic operations [38]; and even after such a pre-processing step, $\kappa$ can still be as large as $d$ in the worst-case. Our algorithm has the benefit of not requiring any pre-processing step.

A third type of algorithms avoid projections by outputting Newton iterates that are guaranteed to be feasible thanks to the use of a self-concordant barrier. The first such algorithm in the context of online learning was introduced by [2]. They presented a general recipe for using self-concordant barriers with Newton steps in online linear optimization. However, their approach falls short of being computationally-efficient as their algorithm needs to compute the inverse of the Hessian of the barrier at every iteration. Inspired by the work of [2], [31] used damped Newton steps with quadratic terms added to the barrier to design an efficient algorithm for the classical portfolio selection problem. Closer to our work is that of [30] who used a similar barrier for designing an algorithm for exp-concave optimization that can be viewed as a computationally-efficient version of the Online Newton Step [13]. Similar to our work, [30] also leverage the stability of the Newton iterates to avoid computing the inverse of the Hessian of the barrier at every step. However, their approach and analysis, which are tailored to the exp-concave setting do not necessarily lead to improved regret bounds in the general OCO setting we consider. In particular, their algorithm does not lead to a $O(\sqrt{T})$ regret bound over polytopes.

Finally, for our application to polytopes, we make use of recent tools and techniques developed for solving linear programs efficiently. In particular, we make use of the Lee-Sidford barrier [20, 21, 22], which can be computed efficiently and, when used to compute Newton iterates, leads to the state-of-the-art $\widetilde{O}(\sqrt{d})$ iteration upper-bound for solving a linear program. For the OCO setting, we show that using the Lee-Sidford barrier within our algorithm leads to a $\widetilde{O}(\sqrt{T})$ regret bound. We also note that ideas similar to the ones we use to avoid computing the inverse of the Hessian of the barrier at every round were used to amortize computations in the context of solving linear programs (see e.g. [4, 39, 40]).

**Outline.** In section 2, we present our notation and relevant definitions. In Section 3, we present our algorithm and guarantees. In Section 4, we apply our results to the case of a polytope. All the proof are differed to the appendix.

## 2    Preliminaries

Throughout the paper, we let $\mathcal{K}$ be a closed convex subset of $\mathbb{R}^d$. We denote by $\|\cdot\|$ the Euclidean norm and by $\mathcal{B}(R) \subset \mathbb{R}^d$ the Euclidean ball of radius $R > 0$. We let $\text{int}\,\mathcal{K}$ denote the interior of $\mathcal{K}$.

Our main algorithm, which can be viewed as an "online" counter-part to the Newton iterations [33], uses a self-concordant barrier over the set of interest to avoid the need of performing Euclidean projections onto $\mathcal{K}$. Next, we present the definition of a self-concordant barrier.

**Self-concordant barriers.** For the rest of this section, we let $\mathcal{K}$ be a convex compact set with non-empty interior $\operatorname{int}\mathcal{K}$. For a twice [resp. thrice] differentiable function, we let $\nabla^2 f(\boldsymbol{u})$ [resp. $\nabla^3 f(\boldsymbol{u})$] be the Hessian [resp. third derivative tensor] of $f$ at $\boldsymbol{u}$.

**Definition 1** (Self-concordant function). *A convex function $f: \operatorname{int}\mathcal{K} \to \mathbb{R}$ is called* self-concordant *with constant $M_f \geq 0$, if $f$ is $C^3$ and satisfies*

- *$f(x_k) \to +\infty$ for $x_k \to x \in \partial\mathcal{K}$; and*

- *For all $x \in \operatorname{int}\mathcal{K}$ and $u \in \mathbb{R}^d$, $|\nabla^3 f(x)[u,u,u]| \leq 2M_f \|u\|^3_{\nabla^2 f(x)}$.*

**Definition 2** (Self-concordant barrier). *For $M_f, \nu \geq 0$, we say that $f: \operatorname{int}\mathcal{K} \to \mathbb{R}$ is a $(M_f, \nu)$-self-concordant barrier for $\mathcal{K}$ if $f$ is a self-concordant function over $\mathcal{K}$ with constant $M_f$ and*

$$\forall w \in \operatorname{int}\mathcal{K}, \quad \nabla f(w)^\top \nabla^{-2} f(w)\nabla f(w) \leq \nu.$$

**Computational Oracles.** We will assume that our algorithm has access to a self-concordant function over the set $\mathcal{K}$ through the following gradient and Hessian Oracles.

**Definition 3** (Gradient Oracle). *Given a point $w \in \operatorname{int}\mathcal{K}$ and a tolerance $\varepsilon > 0$, the gradient Oracle $\mathcal{O}^{\mathrm{grad}}_\varepsilon(\Phi)$ returns an $\varepsilon$-approximate vector $\widehat{\nabla}_w$ of the gradient $\nabla\Phi(w)$ in the dual local norm of the Hessian:*

$$\|\widehat{\nabla}_w - \nabla\Phi(w)\|_{\nabla^{-2}\Phi(w)} \leq \varepsilon.$$

*We denote by $\mathcal{C}^{\mathrm{grad}}_\varepsilon(\Phi)$ the computational cost of one call to $\mathcal{O}^{\mathrm{grad}}_\varepsilon(\Phi)$.*

When clear from the context, we will simply write $\mathcal{C}^{\mathrm{grad}}_\varepsilon$ and $\mathcal{O}^{\mathrm{grad}}_\varepsilon$ for $\mathcal{C}^{\mathrm{grad}}_\varepsilon(\Phi)$ and $\mathcal{O}^{\mathrm{grad}}_\varepsilon(\Phi)$, respectively.

**Definition 4** (Hessian Oracle). *Given a point $w \in \operatorname{int}\mathcal{K}$ and a tolerance $\varepsilon > 0$, the Hessian Oracle $\mathcal{O}^{\mathrm{hess}}_\varepsilon(\Phi)$ returns a matrix $H$ and its inverse $H^{-1}$ which are $1 \pm \varepsilon$ spectral approximations of the Hessian and inverse Hessian of $\Phi$ at $w$:*

$$(1-\varepsilon)\nabla^2\Phi(w) \preccurlyeq H \preccurlyeq (1+\varepsilon)\nabla^2\Phi(w) \quad and \quad (1-\varepsilon)\nabla^{-2}\Phi(w) \preccurlyeq H^{-1} \preccurlyeq (1+\varepsilon)\nabla^{-2}\Phi(w).$$

*We denote by $\mathcal{C}^{\mathrm{hess}}_\varepsilon(\Phi)$ the computational cost of one call to $\mathcal{O}^{\mathrm{hess}}_\varepsilon(\Phi)$.*

When clear from the context, we will simply write $\mathcal{C}^{\mathrm{hess}}_\varepsilon$ and $\mathcal{O}^{\mathrm{hess}}_\varepsilon$ for $\mathcal{C}^{\mathrm{hess}}_\varepsilon(\Phi)$ and $\mathcal{O}^{\mathrm{hess}}_\varepsilon(\Phi)$, respectively.

**Additional notation.** We use the notation $f \lesssim g$ to mean $f \leq Cg$ for some universal constant $C > 0$. We also write $f \leq \widetilde{O}g$ to mean $f \leq \operatorname{polylog}(T, d) \cdot g$. We let $\nabla^{-2} := (\nabla^2)^{-1}$ and $\nabla^{-1/2}$ refer to the inverse of the Hessian and the inverse of the square root of the Hessian, respectively.

## 3 Algorithm and Regret Guarantees

In this section, we construct a projection-free algorithm for Online Convex Optimization. The algorithm in question (Alg. 1) outputs approximate Newton iterates with respect to "potential functions" $(\Phi_t)$ that take the following form:

$$\Phi_t(w) := \Phi(w) + w^\top \sum_{s=1}^{t-1} g_s,$$

where $(g_s \in \partial\ell_s(w_s))$ are the sub-gradients of the losses $(\ell_s)$ at the iterates $(w_s)$ of Algorithm 1, and $\Phi$ is a self-concordant function over $\mathcal{K}$. Algorithm 1 uses the the approximate gradient and Hessian Oracles of $\Phi$ (see 2) to output iterates $(w_t)$ approximate Newton iterates in the following sense:

$$\forall t \in [T], \quad w_{t+1} \approx w_t - \nabla^{-2}\Phi_{t+1}(w_t)\nabla\Phi_{t+1}(w_t). \tag{1}$$

As is by now somewhat standard in the analyses of online Newton iterates of the form in (1), we will bound the regret of Algorithm 1 by showing that:

- The iterates $(w_t)$ are close (in the norm induced by the Hessian $\nabla^2 \Phi(w_t)$) to the FTRL iterates, which are given by

$$w_t^\star \in \operatorname*{argmin}_{w \in \mathcal{K}} \Phi_t(w). \tag{2}$$

- The regret of FTRL is bounded by $O(\sqrt{T})$.

Our main contribution is an algorithm that outputs iterates $(w_t)$ that satisfy the first bullet point (i.e. iterates that satisfy (1)) while only calling a Hessian Oracle (which is potentially computationally expensive) a $O(1/\sqrt{T})$ fraction of the rounds after $T$ rounds. As we show in Section 4, for the case where $\mathcal{K}$ is a polytope with $m \in \mathbb{N}$ constraints, the algorithm achieves a $\widetilde{O}(\sqrt{T})$ regret bound, where the per-iteration computational cost essentially reduces to a linear-system-solve involving a $d \times m$ matrix. Among existing OCO algorithms that achieve a $\widetilde{O}(\sqrt{T})$ regret bound, none can achieve this computational complexity for general polytopes with $m$ constraints (see Section 4 for more details).

### 3.1 Efficient Computation of the Newton Iterates with `BARONS`

The key feature of `BARONS` (Algorithm 1) is that is uses an amortized computation of the Hessians. Namely, `BARONS` computes the inverse of the Hessian of the barrier $\Phi$ only for a small fractions of the iterates $(w_t)$. Henceforth, we refer to the iterates where the algorithm computes the full inverse of the Hessian as *landmark iterates*; these are the iterates $(u_t)$ in Lines 13 and 16 of Algorithm 1. The idea behind this is that for a sufficiently curved[3] barrier $\Phi$, the Newton iterates with respect to $\Phi$ are stable enough that it suffices to compute the inverse of the Hessian of $\Phi$ at the closest landmark iterate. For example, this is what was done in [30] to design an efficient algorithm for exp-concave optimization.

Unlike the setting of [30], where it is possible to add quadratic terms to the barrier for additional stability, in our setting we cannot do that without sacrificing performance in terms of regret. Without the quadratic terms, the Newton iterates are not stable enough for our desired guarantee. Instead of adding regularization terms, `BARONS` takes $\widetilde{O}(1)$ Newton steps per round to get "closer" to the Newton iterate with the true Hessian matrix. This simple approach is key to the success of our approach.

In the next subsection, we give a generic guarantee for `BARONS`.

### 3.2 Generic Regret Guarantee of `BARONS`

In this subsection, we present a general regret and computational guarantee for `BARONS` under minimal assumptions on the sequence of losses and without turning the "step size" $\eta$. In the next subsection, we will instantiate the regret guarantee when additional assumptions on the sequence of losses are available. We now state the main guarantee of `BARONS` (the proof in Appendix C.1).

**Theorem 5** (Master theorem). *Let $\Phi$ be a self-concordant function over $\mathcal{K}$ with constant $M_\Phi > 0$, and let $b, \eta, \varepsilon, \alpha > 0$ and $m_{\texttt{Newton}} \in \mathbb{N}$ be such that $\eta \leq \frac{1}{1000 b M_\Phi}$, $\varepsilon \leq \frac{1}{20000 M_\Phi}$, $\alpha = 0.001$, and $m_{\texttt{Newton}} \coloneqq \Theta(\log \frac{1}{\varepsilon M_\Phi})$. Further, let $(w_t)$ be the iterates of Algorithm 1 with input $(\eta, \varepsilon, \alpha, m_{\texttt{Newton}})$ and suppose that the corresponding sub-gradients $(g_t)$ satisfy $\|g_t\|_{\nabla^{-2}\Phi(w_t)} \leq b$, for all $t \geq 1$. Then, the regret of Algorithm 1 is bounded as:*

$$\sum_{t=1}^{T} (\ell_t(w_t) - \ell_t(w)) \lesssim \frac{1}{\eta} \Phi(w) + \eta \sum_{t=1}^{T} \|g_t\|_{\nabla^{-2}\Phi(w_t)}^2 + \varepsilon \sum_{t=1}^{T} \|g_t\|_{\nabla^{-2}\Phi(w_t)}, \quad \forall w \in \operatorname{int} \mathcal{K}. \tag{3}$$

*Furthermore, the computational cost of the algorithm is bounded by*

$$O\left( (\mathcal{C}_\varepsilon^{\texttt{grad}} + d^2) \cdot T \cdot \log \frac{1}{\varepsilon M_\Phi} + \mathcal{C}_\alpha^{\texttt{hess}} \cdot \left( M_\Phi T \varepsilon + M_\Phi \sum_{t=1}^{T} \eta \|g_t\|_{\nabla^{-2}\Phi(w_t)} \right) \right).$$

Theorem 5 essentially shows that it is possible to achieve the same regret as FTRL, while only computing the inverse of the Hessian of $\Phi$ at most $\widetilde{O}(M_\Phi T \eta)$ number of times.

---

[3]Informally, the "curvature" of a convex function is high when the rate of change of its gradients is high.

---

**Algorithm 1** `BARONS`: Barrier-Regularized Online Newton Step

---

**Require:**

- Parameters $\eta, \varepsilon, \alpha > 0$, and $m_{\texttt{Newton}} \geq 1$.
- Gradient/Hessian Oracles $\mathcal{O}_\varepsilon^{\texttt{grad}}/\mathcal{O}_\alpha^{\texttt{hess}}$ for self-concordant function $\Phi$ with constant $M_\Phi > 0$.
- $w^\star \in \operatorname{argmin}_{w \in \mathcal{K}} \Phi(w)$.

1: Set $u_1 = w_1 = w^\star$, $H_1 = \mathcal{O}_\alpha^{\texttt{hess}}(w^\star)$, and $s_0 = 0$.
2: **for** $t = 1, \ldots, T$ **do**
3:     Play $w_t$ and observe $g_t \in \partial \ell_t(w_t)$.
4:     Set $s_t = s_{t-1} + \eta g_t$.
5:     Set $w_{t+1}^1 = w_t$.
    // Perform intermediate Newton steps to decrease the Newton decrement
6:     **for** $m = 1, \ldots, m_{\texttt{Newton}} - 1$ **do**
7:         Set $\widehat{\nabla}_{t+1}^m = \mathcal{O}_\varepsilon^{\texttt{grad}}(w_{t+1}^m)$.
8:         Set $\widetilde{\nabla}_{t+1}^m = \widehat{\nabla}_{t+1}^m + s_t$.  // $\widetilde{\nabla}_{t+1}^m \approx \nabla \Phi_{t+1}(w_{t+1}^m)$
9:         Set $w_{t+1}^{m+1} = w_{t+1}^m - H_t^{-1} \widetilde{\nabla}_{t+1}^m$.
10:    **end for**
11:    Set $w_{t+1} = w_{t+1}^{m+1}$.
    // Check if the landmark needs updating
12:    **if** $\|w_{t+1} - u_t\|_{H_t} \leq 1/(41 M_\Phi)$ **then**
13:       Set $u_{t+1} = u_t$. // Update landmark
14:       $H_{t+1} = H_t$ and $H_{t+1}^{-1} = H_t^{-1}$.
15:    **else**
16:       Set $u_{t+1} = w_{t+1}$. // No landmark update
17:       Set $H_{t+1} = \mathcal{O}_\alpha^{\texttt{hess}}(u_{t+1})$ and compute $H_{t+1}^{-1}$.
18:    **end if**
19: **end for**

---

## 3.3 Regret Guarantee Under Local and Euclidean Norm Bounds on the Sub-Gradients

We now instantiate the guarantee in Theorem 5 with a $(M_\Phi, \nu)$-self-concordant barrier $\Phi$ for the set $\mathcal{K}$, with respect to which the local norms of the sub-gradients are bounded; that is, when $\|g_t\|_{\nabla^{-2}\Phi(w_t)} \leq b$. We note that the regret bound in (36) has an additive $\Phi(w)$ which may be unbounded near the boundary of $\mathcal{K}$. However, it is still possible to compete against comparators in $\operatorname{int} \mathcal{K}$ by making additional assumptions on the range of the losses [27, 32]. We discuss some of these assumptions in the sequel. For the next theorem, we will state the regret bound of `BARONS` relative to comparators in the restricted set:

$$\mathcal{K}_c \coloneqq (1 - c)\mathcal{K} \oplus \{c w^\star\}, \tag{4}$$

where $\oplus$ denotes the Minkowski sum, $w^\star \in \operatorname{argmin}_{w \in \mathcal{K}} \Phi(w)$, and $c \in (0, 1)$ is a parameter.

With this, we now state a regret bound for `BARONS` when the sub-gradients of the losses have bounded local norms. The proof of the next theorem is in Appendix C.2.

**Theorem 6** (Local norm bound). *Let $\Phi$ be an $(M_\Phi, \nu)$-self-concordant barrier for $\mathcal{K}$ and let $c \in (0, 1), b > 0$. Further, suppose that for all $t \in [T]$, $\|g_t\|_{\nabla^{-2}\Phi(w_t)} \leq b$, where $(w_t)$ are the iterates of `BARONS` with input parameters $(\eta, \varepsilon, \alpha, m_{\texttt{Newton}})$ such that*

$$\eta \coloneqq \sqrt{\frac{\nu \log c}{b^2 T}}, \quad \varepsilon \coloneqq \sqrt{\frac{\nu}{T}}, \quad \alpha \coloneqq 0.001, \quad and \quad m_{\texttt{Newton}} \coloneqq \Theta\left(\log \frac{1}{\varepsilon M_\Phi}\right). \tag{5}$$

*For $T \geq 1$ large enough such that $\eta \leq \frac{1}{1000 b M_\Phi}$, $\varepsilon \leq \frac{1}{20000 M_\Phi}$, the regret of `BARONS` is bounded as*

$$\operatorname{Reg}_T^{\texttt{BARONS}}(w) \lesssim b\sqrt{\nu T \log c}, \quad \forall w \in \mathcal{K}_c, \tag{6}$$

*where $\mathcal{K}_c \subset \mathcal{K}$ is as in (4). Further, the computational complexity of `BARONS` in this case is bounded by*

$$O\left(\left(\mathcal{C}_\varepsilon^{\texttt{grad}} + d^2\right) \cdot T \cdot \log \frac{T}{\nu M_\Phi} + \mathcal{C}_\alpha^{\texttt{hess}} \cdot M_\Phi \sqrt{T \nu \log c}\right).$$

**Remark 1.** *The regret bound in Theorem 6 is stated with respect to comparators in the restricted set $\mathcal{K}_c$ defined in* (4). *It is possible to extend this guarantee to all comparators in* $\operatorname{int} \mathcal{K}$ *under an additional assumption on the range of the losses. For example, if for $w^\star \in \operatorname{argmin}_{w \in \mathcal{K}} \Phi(w)$, we have*

$$\sup_{w \in \operatorname{int} \mathcal{K}, t \in [T]} \ell_t \left( \left(1 - \frac{1}{T}\right) \cdot w + \frac{1}{T} \cdot w^\star \right) - \ell(w) \le O\left(\frac{1}{\sqrt{T}}\right), \tag{7}$$

*then the regret guarantee in* (6) *can be extended to all comparators in* $\operatorname{int} \mathcal{K}$ *up to an additive $O(\sqrt{T})$ term (see Lemma 7 in the appendix). In this case, the $\log c$ term in the computational complexity need be replaced by $\log T$. We note that the condition in* (7) *does not require a uniform bound on the losses. Instead, it only restrict the rate of growth of the losses $(\ell_t(w))$ as $w$ approaches the boundary of $\mathcal{K}$. As we show in the sequel (§4.2),* (7) *is satisfied for some popular losses which are* not *Lipschitz.*

We now instantiate the guarantee in Theorem 5 when the sub-gradients are bounded in Euclidean norm (instead of local norm); that is, we assume that for all $t \in [T]$, $\|g_t\| \le G$ for some $G > 0$. We note that this assumption implies (7), and we will be able to bound the regret against all comparators in $\operatorname{int} \mathcal{K}$ as alluded to in Remark 1. The proof of the next theorem is in Appendix C.3).

**Theorem 7** (Euclidean norm bound)**.** *Let $\Psi$ be an $(M_\Psi, \nu)$ self-concordant barrier for $\mathcal{K}$ and let $\Phi(\cdot) \coloneqq \Psi(\cdot) + \frac{\nu}{2R^2}\|\cdot\|^2$. Further, let $G, R > 0$ and suppose that $\mathcal{K} \subseteq \mathcal{B}(R)$ and for all $t \in [T]$, $\|g_t\| \le G$, where $g_t \in \partial \ell_t(w_t)$ and $(w_t)$ are the iterates of* BARONS *with input parameters $(\eta, \varepsilon, \alpha, m_{\mathtt{Newton}})$ such that*

$$\eta \coloneqq \frac{\nu}{RG} \sqrt{\frac{\log T + 1}{T}}, \quad \varepsilon \coloneqq \sqrt{\frac{\nu}{T}}, \quad \alpha \coloneqq 0.001, \quad and \quad m_{\mathtt{Newton}} \coloneqq \Theta\left(\log \frac{1}{\varepsilon M_\Psi}\right). \tag{8}$$

*For $T \ge 1$ large enough such that $\eta \le \frac{1}{1000 G M_\Psi}$, $\varepsilon \le \frac{1}{20000 M_\Psi}$, the regret of* BARONS *is bounded as*

$$\operatorname{Reg}_T^{\mathtt{BARONS}}(w) \lesssim RG\sqrt{T \log T}, \quad \forall w \in \operatorname{int} \mathcal{K}. \tag{9}$$

*Further, the computational complexity of* BARONS *in this case is bounded by*

$$O\left( \left( \mathcal{C}_\varepsilon^{\mathtt{grad}}(\Psi) + d^2 \right) \cdot T \cdot \log \frac{T}{\nu M_\Psi} + \mathcal{C}_\alpha^{\mathtt{hess}}(\Psi) \cdot M_\Psi \sqrt{T \nu \log T} \right).$$

## 4 Application to Polytopes Using the Lee-Sidford Barrier

In this section, we assume that the set $\mathcal{K}$ is a polytope in $\mathbb{R}^d$ specified by $m$ linear constraints:

$$\mathcal{K} = \{w \in \mathbb{R}^d \mid \forall i \in [m], \ a_i^\top w \ge b_i'\}, \tag{10}$$

and we construct efficient gradient and Hessian Oracles for a self-concordant barrier for $\mathcal{K}$. This will then allow us to instantiate the guarantees of BARONS in Section 3 and provide explicit and state-of-the-art bounds on the regret of BARONS.

We will assume without loss of generality that $\|a_i\| = 1$, for all $i \in [m]$, and let $A \coloneqq (a_1, \dots, a_m)^\top \in \mathbb{R}^{m \times d}$ denote the *constraint* matrix of the set $\mathcal{K}$. For the rest of this section, it will be convenient to define the "slack" variables $s_{w,i} = a_i^\top w - b_i'$, for $i \in [m]$. Here, $s_{w,i}$ essentially represents the distance of $w$ to the $i$th facet of the polytope $\mathcal{K}$. Further, we let $S_w \coloneqq \operatorname{diag}(s_w)$ be the diagonal matrix whose $i$th diagonal entry is $s_{w,i}$.

**The LS barrier.**    To perform Online Convex Optimization over $\mathcal{K}$, we pick the regularizer $\Phi$ of BARONS to be the Lee-Sidford (LS) barrier $\Phi^{\mathtt{LS}}$ [22] with parameter $p > 0$, which is defined as

$$\Phi^{\mathtt{LS}}(v) = \min_{v \in \mathbb{R}_{>0}^m} -\log \det(A^\top S_w V S_w A) + \frac{1}{1 + p^{-1}} \operatorname{tr}(V^{1+1/p}),$$

where $V = \operatorname{diag}(v)$. One way to think of the LS barrier is as a weighted log-barrier. As we will discuss in the sequel, this choice will confer computational and performance (in terms of regret) advantages over the standard log-barrier.

**Self-concordance of the LS barrier.** According to [8, Theorem 30], the LS barrier with the choice $p = O(\log(m))$ is a self-concordant function with parameter $M_{\Phi^{\text{LS}}}$ satisfying

$$M_{\Phi^{\text{LS}}} = O(\log(m)^{2/5}) = \widetilde{O}(1),$$

The other favorable property of this barrier is that its Newton decrement at any point $w \in \mathcal{K}$ is of order $\widetilde{O}(\sqrt{d})$; that is,

$$\|\nabla \Phi^{\text{LS}}(w)\|_{\nabla^{-2}\Phi^{\text{LS}}(w)} = \widetilde{O}(\sqrt{d}). \tag{11}$$

Therefore, $\Phi^{\text{LS}}$ is a $(\widetilde{O}(1), \widetilde{O}(d))$-self-concordant barrier. For the log-barrier, the right-hand side of (11) would be $\sqrt{m}$.

**Cost of gradient and Hessian Oracles.** We consider the computational complexities of gradient and Hessian Oracles for $\Phi^{\text{LS}}$. By [22], we have that for $\varepsilon > 0$,

$$\mathcal{C}_\varepsilon^{\text{grad}}(\Phi^{\text{LS}}) \leq \widetilde{O}(\mathcal{C}^{\text{sys}} \cdot \log(1/\varepsilon)), \quad \text{and} \quad \mathcal{C}_\varepsilon^{\text{hess}}(\Phi^{\text{LS}}) \leq \widetilde{O}(\mathcal{C}^{\text{sys}}\sqrt{d} \cdot \log(1/\varepsilon)),$$

where $\mathcal{C}^{\text{sys}}$ is the computational cost of solving a linear system of the form $A^\top \text{diag}(v) A x = y$, for vectors $v \in \mathbb{R}_{\geq 0}^d$ and $y \in \mathbb{R}^d$; we recall that $A = (a_1, \ldots, a_m)^\top$ is the constraint matrix for $\mathcal{K}$. In the worst-case, such a linear system can be solved with cost bounded as

$$\mathcal{C}^{\text{sys}} \leq O(md^{\omega-1}), \tag{12}$$

where $\omega$ is the exponent of matrix multiplication, and $m$ is the number of constraints of $\mathcal{K}$. However, as we show in the sequel, $\mathcal{C}^{\text{sys}}$ can be much smaller in many practical applications.

With this, we immediately obtain the following corollary for the regret and run-time of BARONS under local norm and Euclidean norm bounds on the sub-gradients.

**Corollary 1** (OCO over a polytope with LS barrier)**.** *Let $c \in (0,1), G, R, b > 0$, and suppose $\mathcal{K}$ is given by (10) and that $\Phi^{\text{LS}}$ is the corresponding LS barrier. Further, let $(w_t)$ be the iterates of BARONS, and let $\mathcal{K}_c$ be the restricted version of $\mathcal{K}$ defined in (4). Then, the following holds:*

- ***Local norm bound:*** *If $\|g_t\|_{\nabla^{-2}\Phi(w_t)} \leq b$, for all $t \geq 1$, and the parameters $(\eta, \varepsilon, \alpha, m_{\text{Newton}})$ of BARONS are set as in Theorem 6 with $\Phi = \Phi^{\text{LS}}$ and $(M_\Phi, \nu) = (\widetilde{O}(1), \widetilde{O}(d))$, then for $T$ large enough (as specified in Theorem 6), the regret of BARONS is bounded by*

$$\text{Reg}_T^{\text{BARONS}}(w) \lesssim b\sqrt{dT \log c}, \quad \forall w \in \mathcal{K}_c. \tag{13}$$

- ***Euclidean norm bound:*** *If $\mathcal{K} \subseteq \mathcal{B}(R)$ and $\|g_t\| \leq G$, for all $t \geq 1$, and the parameters $(\eta, \varepsilon, \alpha, m_{\text{Newton}})$ of BARONS are set as in Theorem 7 with $\Phi(\cdot) = \Phi^{\text{LS}}(\cdot) + \frac{\nu}{2R^2}\|\cdot\|^2$ and $(M_\Psi, \nu) = (\widetilde{O}(1), \widetilde{O}(d))$, then for $T$ large enough (as in Theorem 7) BARONS has regret bounded as*

$$\text{Reg}_T^{\text{BARONS}}(w) \lesssim RG\sqrt{T \log T}, \quad \forall w \in \text{int}\,\mathcal{K}. \tag{14}$$

*In either case, the computational complexity is bounded by*

$$\widetilde{O}\Big((\mathcal{C}^{\text{sys}} + d^2) \cdot T + \mathcal{C}^{\text{sys}} \cdot d\sqrt{T}\Big), \tag{15}$$

*where $\mathcal{C}^{\text{sys}}$ is the computational cost of solving a linear system of the form $A^\top \text{diag}(v) A x = y$, for vectors $v \in \mathbb{R}_{\geq 0}^d$ and $y \in \mathbb{R}^d$ (recall that $A$ is the constraint matrix for the polytope $\mathcal{K}$).*

**Using the log-barrier.** We note that since $\mathcal{K}$ is a polytope, we could have used the standard log-barrier

$$\Phi^{\log}(w) \coloneqq \sum_{i=1}^m \log(b_i' - a_i^\top w). \tag{16}$$

This barrier is $(1, m)$-self-concordant, and so instantiating Theorem 5 with it would imply a $\widetilde{O}(b\sqrt{mdT})$ regret bound in the case of local sub-gradient norms bounded by $b > 0$. Using the LS barrier replaces the $\sqrt{m}$ term in this bound by $\sqrt{d}$ regardless of the number of constraints—see

(13). However, this comes at a $\mathcal{C}^{\texttt{sys}}$ computational cost, which can be as high as $md^{\omega-1}$ in the worst-case (see (12)). In the case of the log-barrier, this cost would be replaced by $md$ (essentially because $\mathcal{C}_{\varepsilon}^{\texttt{grad}}(\Phi^{\log}) \leq O(md)$). Thus, when $m$ is of the order of $d$, using the log-barrier may be more computational-efficient compared to using the LS barrier. In the next corollary, we bound the regret of BARONS when $\Phi = \Phi^{\log}$; this result is an immediate consequence of Theorem 7.

**Corollary 2** (OCO over a polytope with the **log** barrier)**.** *Let $G, b > 0$, and suppose $\mathcal{K}$ is given by* (10) *and that $\Phi^{\log}$ is the corresponding* log*-barrier. Further, let $(w_t)$ be the iterates of* BARONS. *If $\mathcal{K} \subseteq \mathcal{B}(R)$ and $\|g_t\| \leq G$, for all $t \geq 1$, and the parameters $(\eta, \varepsilon, \alpha, m_{\texttt{Newton}})$ of* BARONS *are set as in Theorem 7 with $\Phi(\cdot) = \Phi^{\log}(\cdot) + \frac{\nu}{2R^2}\|\cdot\|^2$ and $(M_\Psi, \nu) = (1, m)$, then for $T$ large enough (as in Theorem 7)* BARONS *has regret bounded as*

$$\operatorname{Reg}_T^{\texttt{BARONS}}(w) \lesssim RG\sqrt{T\log T}, \quad \forall w \in \operatorname{int}\mathcal{K}. \tag{17}$$

*The computational complexity is bounded by*

$$\widetilde{O}\Big((md + d^2) \cdot T + md^{\omega-1}\sqrt{mT}\Big). \tag{18}$$

## 4.1 Implications for Lipschitz Losses

We now discuss implications of Corollary 1, and compare the bound of BARONS to those of existing algorithms for Lipschitz losses.

**Dimension-free regret bound.** We note when the Euclidean norms of the sub-gradients are bounded, BARONS achieves a *dimension-free* $O(\sqrt{T})$ regret bound. In contrast, the best dimension-free regret bound[4] achieved by existing projection-free algorithms is of order $O(T^{3/4})$ (see e.g. [14, 11]). We also note that existing separation/membership-based algorithms that achieve a $\sqrt{T}$ regret; for examples those presented in [28, 11, 26], are not dimension-free. Their regret bounds are of order $O(\kappa\sqrt{T})$, where $\kappa = R/r$ with $r, R > 0$ such that $\mathcal{B}(r) \subseteq \mathcal{K} \subseteq \mathcal{B}(R)$. The asphericity parameter can depend on the dimension $d$ [28], and even after a pre-conditioning step (which would involve putting the set $\mathcal{K}$ into near-isotropic position and can cost up to $\Omega(d^4)$ [25]), $\kappa$ can be as large as $d$ in the worst-case. Of course, to make a fair comparison with existing projection-free algorithms, we also need to take computational complexity into account. This is what we do next.

**Computational cost.** The computational cost in (15) should be compared with that of existing projection-free algorithms. For linear optimization-based projection-free algorithms, the computational cost after $T$ rounds is typically of order $\mathcal{C}^{\texttt{lin}} \cdot T$, where $\mathcal{C}^{\texttt{lin}}$ is the cost of performing linear optimization over $\mathcal{K}$ which, for a polytope $\mathcal{K}$, reduces to solving a linear program. Using state-of-the-art interior point methods for solving such a linear program would cost $\mathcal{C}^{\texttt{lin}} \leq \widetilde{O}(\sqrt{d} \cdot \mathcal{C}^{\texttt{sys}})$; see e.g. [22]. Thus, linear optimization-based projection-free algorithms[5] can have a cost that is a factor $\sqrt{d}$ worse than that of BARONS in the setting of Corollary 1. On the other hand, separation/membership-based algorithms, the computational cost scales with $O(\mathcal{C}^{\texttt{sep}} \cdot T)$ after $T$ rounds, where $\mathcal{C}^{\texttt{sep}}$ is the cost of performing separation for the set $\mathcal{K}$. For a general polytope in $\mathbb{R}^d$ with $m$ constraints, we have $\mathcal{C}^{\texttt{sep}} \leq O(md)$, which may be smaller than $\mathcal{C}^{\texttt{sys}}$ (the latter can be as large as $md^{\omega-1}$ in the worse case; see (12)). Here, it may be more appropriate to compare against the computational guarantee of BARONS given in Corollary 2; by (18), we have that for $T \geq d^{\omega-2}\sqrt{m}$, the computational cost of BARONS in the setting of the corollary is dominated by $(md + d^2) \cdot T$, which is comparable to that of existing separation-based algorithms.

## 4.2 Implications for Non-Lipschitz Losses

Another advantage BARONS has over projection-free, and even projection-based, algorithms is that it has a regret bound that scales with a bound on the local norms of the gradients—see (13). We now showcase two online learning settings where this leads to non-trivial performance and computational improvements over existing OCO algorithms.

---

[4]The dependence in $T$ can be improved under additional structure such as smoothness or strong-convexity of the losses.

[5]This only concerns algorithms that use an interior point method to implement linear optimization over $\mathcal{K}$.

**Online Portfolio Selection [5].** The portfolio selection problem is a classical online learning problem where the gradients of the losses can be unbounded. In this paragraph, we demonstrate how the guarantee of BARONS in Corollary 1 leads to a non-trivial guarantee for this setting both in terms of regret and computational complexity. In the online portfolio setting, at each round $t$, a learner (algorithm) chooses a distribution $w_t \in \Delta_d$ over a fixed set of $d$ portfolios. Then, the environment reveals a return vector $r_t \in \mathbb{R}^d_{\geq 0}$, and the learner suffers a loss

$$\ell_t(w_t) \coloneqq -\log w_t^\top r_t.$$

The goal of the learner is to minimize the regret $\operatorname{Reg}_T(w) \coloneqq \sum_{t=1}^T (\ell_t(w_t) - \ell_t(w))$ after $T \geq 1$ rounds. For this problem, it is known that a logarithmic regret is achievable, but the specialized algorithms that achieve this have a computational complexity that scales with $\min(d^3 T, d^2 T^2)$ [5, 27, 32, 41, 17]. On the other hand, applying the generic Online Gradient Descent or the Online Newton Step to this problem leads to regret bounds that scale with the maximum norm of the gradient (which can be unbounded). Instantiating the guarantees of BARONS in Corollary 1 with $\Phi$ set to the standard log-barrier for the simplex[6], in particular the bound in (13), to the online portfolio selection problem leads to an $\widetilde{O}(\sqrt{dT})$ regret bound, which does not depend on the norm of the observed gradients. Furthermore, we have $\mathcal{C}^{\text{sys}} \leq O(d)$, and so by (15) the computational complexity is essentially $O(d^2 T)$ after $T$ rounds. Technically, the bound in (13) is only against comparators in the restricted set $\mathcal{K}_c$. However, by setting $c = 1/T$, it possible to extend this guarantee to all comparators in $\operatorname{int} \mathcal{K}$ as explained in Remark 1, since the losses in this case satisfy (6) [27, Lemma 10].

**Linear prediction with the log-loss.** Another classical online learning problem with unbounded gradients is that of linear prediction with the log-loss [36]. For this problem, at each round $t$, the learner receives a feature vector $x_t \in \mathcal{X} \subseteq \mathbb{R}^d$, outputs $w_t \in \mathcal{W} \subseteq \mathbb{R}^d$, then observes label $y_t \in \mathcal{Y} \coloneqq \{-1, 1\}$ and suffers loss

$$\ell_t(w_t) \coloneqq -\mathbb{I}\{y_t = 1\} \cdot \log(1 - w_t^\top x_t) - \mathbb{I}\{y_t = 0\} \cdot \log(1 - w_t^\top x_t).$$

In the settings, where $(\mathcal{X}, \mathcal{W}) = (\Delta_d, \mathcal{B}_\infty(1))$ and $(\mathcal{X}, \mathcal{W}) = (\mathcal{B}_\infty(1), \Delta_d)$, we have that $\|\nabla \ell_t(w)\|_{\nabla^{-2}\Phi(w)} \leq O(1)$ for all $w \in \operatorname{int} \mathcal{W}$, where $\Phi$ is set to the corresponding log-barrier for $\mathcal{W}$. Thus, instantiating Corollary 1 (in particular (13)) in this setting implies that BARONS achieves a regret bound of the form:

$$\widetilde{O}(\sqrt{dT}), \tag{19}$$

and has computational complexity bounded by $\widetilde{O}(d^2 T)$, as long as $T \geq d$. Again, we emphasize that the bound in (19) does not depend on the norm of the gradients, which may be unbounded.

Finally, we note that there exist a few specialized algorithms that provide sublinear regret bounds for non-lipschitz losses. This includes, for example, the Soft-Bayes algorithm [34]. However, this algorithm is specialized to the log-loss with a particular dependence on the predictions, and it is not clear, for example, what regret bound it would have in the linear prediction setting and other similar settings with non-Lipschitz losses.

---

[6]Technically, we need to use a barrier for the set $\{\tilde{w} \in \mathbb{R}^d_{\geq 0} \mid \sum_{i \in [d-1]} \tilde{w}_i \leq 1\}$; see e.g. [32].

## Acknowledgments and Disclosure of Funding

ZM acknowledges support from the ONR through awards N00014-20-1-2336 and N00014-20-1-2394.

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

# A  Self-concordance properties

Throughout, for a twice-differentiable function $f\colon \operatorname{int}\mathcal{K} \to \mathbb{R}$, we let $\lambda(x,f) \coloneqq \|\nabla f(x)\|_{\nabla^{-2}f(x)}$ denote the *Newton decrement* of $f$ at $x \in \operatorname{int}\mathcal{K}$.

**Lemma 1.** *Let* $f\colon \operatorname{int}\mathcal{K} \to \mathbb{R}$ *be a self-concordant function with constant* $M_f \geq 1$. *Further, let* $x \in \operatorname{int}\mathcal{K}$ *and* $x_f \in \operatorname{argmin}_{x\in\mathcal{K}} f(x)$. *Then,* **I)** *whenever* $\lambda(x,f) < 1/M_f$, *we have*

$$\|x - x_f\|_{\nabla^2 f(x_f)} \vee \|x - x_f\|_{\nabla^2 f(x)} \leq \lambda(x,f)/(1 - M_f\lambda(x,f));$$

*and* **II)** *for any* $M \geq M_f$, *the* Newton step $x^+ \coloneqq x - \nabla^{-2}f(x)\nabla f(x)$ *satisfies* $x^+ \in \operatorname{int}\mathcal{K}$ *and* $\lambda(x^+,f) \leq M\lambda(x,f)^2/(1 - M\lambda(x,f))^2$.

**Lemma 2.** *Let* $f\colon \operatorname{int}\mathcal{K} \to \mathbb{R}$ *be a self-concordant function with constant* $M_f$ *and* $x \in \operatorname{int}\mathcal{K}$. *Then, for any* $w$ *such that* $r \coloneqq \|w - x\|_{\nabla^2 f(x)} < 1/M_f$, *we have*

$$(1 - M_f r)^2\nabla^2 f(w) \preceq \nabla^2 f(x) \preceq (1 - M_f r)^{-2}\nabla^2 f(x).$$

The following result from [33, Theorem 5.1.5] will be useful to show that the iterates of algorithms are always in the feasible set.

**Lemma 3.** *Let* $f\colon \operatorname{int}\mathcal{K} \to \mathbb{R}$ *be a self-concordant function with constant* $M_f \geq 1$ *and* $x \in \operatorname{int}\mathcal{K}$. *Then,* $\mathcal{E}_x \coloneqq \{w \in \mathbb{R}^d\colon \|w - x\|_x < 1/M_f\} \subseteq \operatorname{int}\mathcal{K}$. *Furthermore, for all* $w \in \mathcal{E}_x$, *we have*

$$\|w - x\|_w \leq \frac{\|w - x\|_x}{1 - M_f\|w - x\|_x}.$$

Finally, we will also make use of the following result due to [31]:

**Lemma 4.** *Let* $f\colon \operatorname{int}\mathcal{K} \to \mathbb{R}$ *be a self-concordant function with constant* $M_f > 0$. *Then, for any* $x, w \in \operatorname{int}\mathcal{K}$ *such that* $r \coloneqq \|x - w\|_{\nabla^2 f(x)} < 1/M_f$, *we have*

$$\|\nabla f(x) - \nabla f(w)\|_{\nabla^{-2}f(x)}^2 \leq \frac{1}{(1 - M_f r)^2}\|w - x\|_{\nabla^2 f(x)}^2.$$

# B  Technical Lemmas

Our analysis relies on the crucial fact that the Newton decrement can be sufficiently decreased by taking a Newton step using only approximate gradients and Hessians. We state this fact next; the proof is in §D.1.

**Lemma 5** (Decrease in the Newton decrement). *Let* $\Phi$ *be a self-concordant function over* $\operatorname{int}\mathcal{K}$ *with constant* $M_\Phi > 0$, *and let* $y \in \mathbb{R}^d$ *be such that* $\lambda(y,\Phi) \leq 1/(40M_\Phi)$. *Further, let* $H \in \mathbb{R}^{d\times d}$ *and* $\widehat{\nabla}_y \in \mathbb{R}^d$ *be such that*

$$\|\widehat{\nabla}_y - \nabla\Phi(y)\|_{\nabla^{-2}\Phi(y)} \leq \varepsilon < \frac{1}{40M_\Phi}, \tag{20}$$

$$(1 - \alpha)\nabla^2\Phi(y) \preceq H \preceq (1 + \alpha)\nabla^2\Phi(y), \tag{21}$$

*for* $\alpha < 1/5$. *Then, for* $\tilde{y}^+ \coloneqq y - H^{-1}\nabla\Phi(y)$ *and* $y^+ \coloneqq y - H^{-1}\widehat{\nabla}_y$, *we have*

$$\lambda(\tilde{y}^+, \Phi) \leq 9M_\Phi\lambda(y,\Phi)^2 + 2.5\alpha\lambda(y,\Phi),$$
$$\lambda(y^+, \Phi) \leq 20(1 + \alpha)\varepsilon + (1 + 20(1 + \alpha)\varepsilon)\cdot\lambda(\tilde{y}^+, \Phi).$$

Next, we show that as long as the Newton decrement is small enough at the current iterate $w_{t-1}$, the "intermediate" Newton iterates $(w_t^m)$ remain close to the landmark point $u_{t-1}$; this will be important for the proof of Theorem 5. The proof is in §D.2.

**Lemma 6** (Invariance under Newton iterations). *Let* $\Phi$ *be a self-concordant function over* $\operatorname{int}\mathcal{K}$ *with constant* $M_\Phi > 0$. *Let* $b > 0$, $m_{\texttt{Newton}} \coloneqq \Theta(\log\frac{1}{\varepsilon M_\Phi})$, $\alpha \leq 1/(1000M_\Phi)$, $\varepsilon < 1/(20000M_\Phi)$, $\alpha = 0.001$, *and* $\eta \leq 1/(1000M_\Phi b)$. *Further, let* $(w_t, w_t^m, u_t, g_t, H_t)$ *be as in Algorithm 1 with input* $(\eta, \varepsilon, \alpha, m_{\texttt{Newton}})$. *Suppose that at round* $t - 1$ *of Algorithm 1, we have*

$$\lambda(w_{t-1}, \Phi_{t-1}) \leq \alpha \quad \textit{and} \quad \|u_{t-1} - w_{t-1}\|_{\nabla^2\Phi(u_{t-1})} \leq \frac{1}{40M_\Phi}. \tag{22}$$

*For $t > 1$, if the sub-gradient $g_{t-1}$ at round $t-1$ satisfies $\|g_{t-1}\|_{\nabla^{-2}\Phi(w_{t-1})} \le b$, then*

$$\lambda(w_t^m, \Phi_t) \le \left(\frac{15}{16}\right)^{m-1} \lambda(w_t^1, \Phi_t) + 500\varepsilon \le \frac{1}{40M_\Phi}. \tag{23}$$

*Furthermore, we have for all $m \in [m_{\texttt{Newton}}]$:*

$$\frac{1}{2}\nabla^2\Phi(w_t^m) \preccurlyeq \nabla^2\Phi(w_t^\star) \preccurlyeq 2\nabla^2\Phi(w_t^m), \tag{24}$$

$$\|w_t^m - u_{t-1}\|_{H_{t-1}} \le \frac{1}{10M_\Phi}, \tag{25}$$

$$\|w_t^m - w_t^\star\|_{\nabla^2\Phi(w_t^\star)} \le \frac{1}{49M_\Phi}\left(\frac{15}{16}\right)^{m-1} + 240\varepsilon, \tag{26}$$

$$\left|\|w_t^m - u_{t-1}\|_{H_{t-1}} - \|w_{t-1} - u_{t-1}\|_{H_{t-1}}\right| \le 2\eta\|g_{t-1}\|_{\nabla^2\Phi(w_{t-1})} + \frac{1}{40M_\Phi}\left(\frac{15}{16}\right)^{m-1} + 500\varepsilon + 2\alpha, \tag{27}$$

*where $w_t^\star \in \operatorname{argmin}_{w \in \mathcal{K}} \Phi_t(w)$ is the optimum solution of $\Phi_t$.*

We note that we have not made an attempt to optimize over the constants in Lemma 6.

**Lemma 7.** *Let $\Phi$ be a $(M_\Phi, \nu)$-self-concordant barrier for $\mathcal{K}$, and let $w^\star \in \operatorname{argmin}_{w \in \mathcal{K}} \Phi(w)$. Further, suppose that the losses satisfy:*

$$\sup_{w \in \operatorname{int}\mathcal{K}, t \in [T]} \ell_t\left(\left(1 - \frac{1}{T}\right) \cdot w + \frac{1}{T} \cdot w^\star\right) - \ell(w) \le O\left(\frac{1}{\sqrt{T}}\right), \tag{28}$$

*Then, for any $w \in \mathcal{K}$, there exists $\tilde{w} \in \mathcal{K}_{1/T}$ (where $\mathcal{K}_c$ is as in (4)) such that*

$$\sum_{t=1}^T (\ell_t(w_t) - \ell_t(w)) \le \sum_{t=1}^T (\ell_t(w_t) - \ell_t(\tilde{w}) + O(\sqrt{T}). \tag{29}$$

**Proof.** Fix $w \in \mathcal{K}$ and define $\tilde{w} = \frac{1}{T}w^\star + \left(1 - \frac{1}{T}\right) w \in \mathcal{K}_{1/T}$. Then, by (28), we have, for all $t \in [T]$,

$$\ell_t(w_t) - \ell_t(w) \le \ell_t(\tilde{w}_t) - \ell_t(\tilde{w}) + O(T^{-1/2}). \tag{30}$$

Summing this over $t = 1, \ldots, T$ leads to the desired result. $\qquad\square$

## C   Proofs of the Main Results

Next, we present the proof of Theorem 5.

### C.1   Proof of Theorem 5

**Proof.** The proof consists of three parts: I) First, we show that BARONS keeps the Newton decrements $\lambda(w_t, \Phi_t), t \ge 1$, small—this is the main invariant of BARONS; II) Then, we bound the regret of BARONS using this invariant and the results of Lemma 6; III) Finally, we bound the runtime of BARONS.

**Bounding the Newton decrements.** We will show that the Newton decrements satisfy

$$\lambda(w_s, \Phi_s) \le \alpha := \min\left\{\frac{1}{1000M_\Phi}, 1000\varepsilon\right\}, \tag{31}$$

for all $s \ge 1$. We will show (31) by induction over $t \ge 1$.

**Base case.** The base case follows by the facts that $w_1 \in \operatorname{argmin}_{w \in \mathcal{K}} \Phi(w)$, $\Phi_1 \equiv \Phi$ and that the Newton decrement is zero at the minimizer.

**Induction step.** Suppose that (31) holds with $s = t - 1$ for some $t \ge 1$. We will show that it holds for $s = t$. First, note that by the update rule of landmark (see Lines 12 and 17 of Alg. 1), we have that

$$\|w_{t-1} - u_{t-1}\|_{H_{t-1}} \le \frac{1}{41M_\Phi},$$

where $H_{t-1} = \mathcal{O}_\alpha^{\mathtt{hess}}(u_{t-1})$. Thus, by the choice of $\alpha$ in the theorem's statement, we have

$$\|u_{t-1} - w_{t-1}\|_{\nabla^2 \Phi(u_{t-1})} \le \frac{1}{40 M_\Phi}. \tag{32}$$

This, combined with the fact that (31) holds with $s = t - 1$ (the induction hypothesis) implies that the conditions of Lemma 6 are satisfied. This in turn implies

$$\lambda(w_t, \Phi_t) \stackrel{(a)}{=} \lambda(w_t^{m_{\mathtt{Newton}}}, \Phi_t) \le \left(\frac{15}{16}\right)^{m_{\mathtt{Newton}}} \lambda(w_t^1, \Phi_t) + 500\varepsilon \le \frac{50}{M_\Phi}\left(\frac{15}{16}\right)^{m_{\mathtt{Newton}}} + 500\varepsilon \stackrel{(b)}{\le} \alpha,$$

where $(a)$ follows by the fact that $w_t = w_t^{m_{\mathtt{Newton}}}$ (by definition; see Algorithm 1) and $(b)$ follows by the choice of $m_{\mathtt{Newton}}$ in the theorem's statement. This shows that (31) holds for $s = t$ and concludes the induction.

**Bounding the regret.** To bound the regret of $\mathtt{BARONS}$, we make use of the FTRL iterates $\{w_t^\star\}$, which are given by $w_t^\star \in \operatorname{argmin}_{w \in \mathcal{K}} \Phi_t(w)$: By Lemma 6, we have that for all $t \in [T]$,

$$\|w_t - w_t^\star\|_{\nabla^2 \Phi(w_t^\star)} = \|w_t^{m_{\mathtt{Newton}}} - w_t^\star\|_{\nabla^2 \Phi(w_t^\star)} \le \frac{1}{49 M_\Phi}\left(\frac{15}{16}\right)^{m_{\mathtt{Newton}}} + 240\varepsilon = O(\varepsilon), \tag{33}$$

where the last inequality follows by the choice of $m_{\mathtt{Newton}} = \Theta(\log(1/(\varepsilon M_\Phi)))$ in the theorem's statement. Using this and Hölder's inequality, we now bound the sum of linearized losses of the algorithm in terms of the sum of linearized losses with respect to $\{w_t^\star\}$:

$$\sum_{t=1}^T \langle w_t, g_t \rangle \le \sum_{t=1}^T \langle w_t^\star, g_t \rangle + \sum_{t=1}^T \|w_t^\star - w_t\|_{\nabla^2 \Phi(w_t^\star)} \cdot \|g_t\|_{\nabla^2 \Phi(w_t^\star)^{-1}}. \tag{34}$$

Now, by (24) in Lemma 6 (which holds due to (32) and (31) with $s = t - 1$ as we showed in the prequel), we have $\frac{1}{2}\nabla^2 \Phi(w_t) \preccurlyeq \nabla^2 \Phi(w_t^\star)$, which implies that $\|g_t\|_{\nabla^{-2}\Phi(w_t^\star)} \le 2\|g_t\|_{\nabla^{-2}\Phi(w_t)}$. Combining this with (33) and (34), we get that

$$\sum_{t=1}^T \langle w_t, g_t \rangle \le \sum_{t=1}^T \langle w_t^\star, g_t \rangle + O(\varepsilon) \sum_{t=1}^T \|g_t\|_{\nabla^{-2}\Phi(w_t)}. \tag{35}$$

Now fix $w \in \mathcal{K}$. Subtracting $\sum_{t=1}^T \langle w, g_t \rangle$ from both sides of (35) implies the following bound the regret of $\mathtt{BARONS}$:

$$\operatorname{Reg}_T^{\mathtt{BARONS}}(w) \le \sum_{t=1}^T \langle w_t^\star, g_t \rangle - \sum_{t=1}^T \langle w, g_t \rangle + O(\varepsilon) \sum_{t=1}^T \|g_t\|_{\nabla^{-2}\Phi(w_t)},$$

$$\le \frac{1}{\eta}\Phi(w) + \eta \sum_{t=1}^T \|g_t\|_{\nabla^{-2}\Phi(w_t)}^2 + O(\varepsilon) \sum_{t=1}^T \|g_t\|_{\nabla^{-2}\Phi(w_t)},$$

where the last inequality follows by the regret bound of FTRL (see e.g. [30]).

**Bounding the run-time.** Note that $\mathtt{BARONS}$ updates the landmark points on the rounds where $\|u_t - w_t\|_{H_t} > 1/(41 M_\Phi)$. Now, by (27) in Lemma 6 (which holds due to (32) and (31) with $s = t - 1$ as we showed in the prequel), we have

$$\left|\|u_t - w_t\|_{H_t} - \|u_t - w_{t-1}\|_{H_t}\right| = \left|\|u_t - w_t^{m_{\mathtt{Newton}}}\|_{H_t} - \|u_t - w_{t-1}\|_{H_t}\right|,$$

$$\le 2\eta\|g_{t-1}\|_{\nabla^{-2}\Phi(w_{t-1})} + \frac{1}{40 M_\Phi}\left(\frac{15}{16}\right)^{m_{\mathtt{Newton}}} + 500\varepsilon + 2\alpha$$

$$\le 2\eta\|g_{t-1}\|_{\nabla^{-2}\Phi(w_{t-1})} + O(\varepsilon),$$

where the last inequality follows by the choice of $m_{\mathtt{Newton}}$ in the theorem's statement. Hence, the quantity $\|u_t - w_t\|_{H_t}$ increases each time by at most $2\eta\|g_{t-1}\|_{\nabla^{-2}\Phi(w_{t-1})} + O(\varepsilon)$. Therefore, the number of times that the landmark $u_t$ changes is bounded by

$$O\left(\frac{\sum_{t=1}^T (2\eta\|g_t\|_{\nabla^{-2}\Phi(w_t)} + O(\varepsilon))}{1/(41 M_\Phi)}\right) = O\left(M_\Phi T \varepsilon + M_\Phi \sum_{t=1}^T \eta\|g_t\|_{\nabla^{-2}\Phi(w_t)}\right).$$

Thus, the overall computational cost of recalculating the Hessians and their inverses at the landmark iterates is bounded by

$$\mathcal{C}_\alpha^{\mathtt{hess}} \cdot \left( M_\Phi T \varepsilon + M_\Phi \sum_{t=1}^T \eta \|g_t\|_{\nabla^{-2}\Phi(w_t)} \right).$$

where the multiplicative cost $\mathcal{C}_\alpha^{\mathtt{hess}}$ reflects the fact that the instance of BARONS in the theorem's statement needs $1 \pm \alpha$ accurate approximations of the Hessians and their inverses (in the sense of Definition 4) at the landmark iterates. Moreover, BARONS needs to compute an $\varepsilon$-approximate gradient of $\Phi$ at every point $w_t^m$ for all $t \in [T]$ and $m \in [m_{\mathtt{Newton}}]$. Thus, the cost of computing the gradients is $\mathcal{C}_\varepsilon^{\mathtt{grad}} \cdot T \log \frac{1}{\varepsilon M_\Phi}$. Finally, the matrix-vector product $H_t^{-1}\nabla\Phi_t(w)$ in BARONS costs $O(d^2)$ work, and so overall the computational cost is

$$O\left( (\mathcal{C}_\varepsilon^{\mathtt{grad}} + d^2) \cdot T \log \frac{1}{\varepsilon M_\Phi} + \mathcal{C}_\alpha^{\mathtt{hess}} \cdot \left( M_\Phi T \varepsilon + M_\Phi \sum_{t=1}^T \eta \|g_t\|_{\nabla^{-2}\Phi(w_t)} \right) \right).$$

$\square$

## C.2 Proof of Theorem 6

**Proof.** Note that without having any effect on the algorithm, we can add an arbitrary constant to the barrier $\Phi$. Thus, without loss of generality, we assume $\Phi(w^\star) = 0$, which implies $\Phi(w) \geq 0$, for all $w \in \mathcal{K}$. We define the restricted comparator class

$$\widetilde{\mathcal{K}} := \{w \in \mathcal{K} : \Phi(w) \leq \Phi(w^\star) + \nu \log c\}.$$

By [33, Corollary 5.3.3] and the fact that $\Phi$ is an $(M_\Phi, \nu)$-self-concordant barrier for $\mathcal{K}$, we have that

$$\mathcal{K}_c \subseteq \widetilde{\mathcal{K}}, \tag{36}$$

and so it suffices to bound the regret against comparators in $\widetilde{\mathcal{K}}$. Fix $\tilde{w} \in \widetilde{\mathcal{K}}$. Under the assumptions of the theorem, the preconditions of Theorem 5 are satisfied and so we have,

$$\sum_{t=1}^T (\ell_t(w_t) - \ell_t(\tilde{w})) \lesssim \frac{1}{\eta}\Phi(\tilde{w}) + \eta \sum_{t=1}^T \|g_t\|_{\nabla^{-2}\Phi(w_t)}^2 + \varepsilon \sum_{t=1}^T \|g_t\|_{\nabla^{-2}\Phi(w_t)},$$

$$= \frac{1}{\eta}\nu \log c + \eta b^2 T + \varepsilon T b, \quad (\text{since } \tilde{w} \in \widetilde{\mathcal{K}} \text{ and } \|g_t\|_{\nabla^{-2}\Phi(w_t)} \leq b)$$

$$= 2b\sqrt{\nu T \log c} + b\sqrt{\nu T},$$

where in the last step we used the choices of $\eta$ and $\varepsilon$ in (5). Combining this with (36) implies the desired regret bound. The bound on the computational complexity follows immediately from Theorem 5, the fact that $\|g_t\|_{\nabla^{-2}\Phi(w_t)} \leq b$, and the choices of $\eta$ and $\varepsilon$ in (5). $\square$

## C.3 Proof of Theorem 7

**Proof.** Similar to the proof of Theorem 6, and without loss of generality, we assume that $\Psi$ is zero at its minimum, i.e. $\Psi(w^\star) = 0$. We define the restricted comparator class

$$\widetilde{\mathcal{K}} := \{w \in \mathcal{K} : \Psi(w) \leq \Psi(w^\star) + \nu \log T\}. \tag{37}$$

By [33, Corollary 5.3.3] and the fact that $\Psi$ is an $(M_\Psi, \nu)$-self-concordant barrier for $\mathcal{K}$, we have that

$$\mathcal{K}_{1/T} \subseteq \widetilde{\mathcal{K}}. \tag{38}$$

On the other hand, by Lemma 7 we have that

$$\sup_{w \in \mathrm{int}\,\mathcal{K}} \sum_{t=1}^T (\ell_t(w_t) - \ell_t(w)) \leq \sup_{\tilde{w} \in \mathrm{int}\,\mathcal{K}} \sum_{t=1}^T (\ell_t(w_t) - \ell_t(\tilde{w}) + O(\sqrt{T}).$$

Combining this with (38) implies that it suffices to bound the regret against comparators in $\widetilde{\mathcal{K}}$. Fix $\tilde{w} \in \widetilde{\mathcal{K}}$. Note that since $\Phi$ is equal to $\Psi$ plus a quadratic, $\Phi$ is also a self-concordant function with

constant $M_\Phi = M_\Psi$ [33]. Thus, under the assumptions of the theorem the preconditions of Theorem 5 are satisfied and so we have,

$$\sum_{t=1}^{T}(\ell_t(w_t) - \ell_t(\tilde{w})) \lesssim \frac{1}{\eta}\Phi(\tilde{w}) + \eta\sum_{t=1}^{T}\|g_t\|_{\nabla^{-2}\Phi(w_t)}^2 + \varepsilon\sum_{t=1}^{T}\|g_t\|_{\nabla^{-2}\Phi(w_t)}. \tag{39}$$

Now, by the choice of $\Phi$, we have that

$$\|g_t\|_{\nabla^{-2}\Phi(w_t)} \le R\|g_t\|/\sqrt{\nu} \le RG/\sqrt{\nu}.$$

Moreover, from the condition that $\mathcal{K} \subseteq \mathcal{B}(R)$, (37) and the fact that $\Psi(w^\star) = 0$, we have for all $w \in \widetilde{\mathcal{K}}$:

$$\Phi(w) \le \nu\log T + \frac{\nu}{2}.$$

Plugging this into (39) and using that $\tilde{w} \in \mathcal{K}$, we get

$$\sum_{t=1}^{T}(\ell_t(w_t) - \ell_t(\tilde{w})) = \frac{1}{\eta}\nu\log T + \frac{1}{2\eta} + \eta\frac{R^2G^2}{\nu}T + \varepsilon\frac{TRG}{\sqrt{\nu}},$$

$$= 2RG\sqrt{T\log T} + \frac{5}{2}RG\sqrt{T},$$

where in the last step we used the choices of $\eta$ and $\varepsilon$ in (8). Combining this with (38) implies the desired regret bound. The bound on the computational complexity follows from the computational complexity in Theorem 5 and the fact a gradient Oracle $\mathcal{O}_\varepsilon^{\text{grad}}(\Phi)$ [resp. Hessian Oracle $\mathcal{O}_\alpha^{\text{grad}}(\Psi)$] for $\Phi(\cdot) = \Psi(\cdot) + \frac{\nu}{2R^2}\|\cdot\|^2$ can be implemented with one call to $\mathcal{O}_\varepsilon^{\text{grad}}(\Psi)$ [resp. $\mathcal{O}_\alpha^{\text{hess}}(\Psi)$] plus $d$ arithmetic operations. $\qquad\square$

# D  Proofs of the Technical Lemmas

## D.1  Proof of Lemma 5

**Proof.** Throughout, we let $h$ is the Newton step based on the exact gradient $\nabla\Phi(y)$:

$$h = -H^{-1}\nabla\Phi(y).$$

Recall that $\tilde{y}^+$ and $y^+$ from the lemma's statement satisfy

$$\tilde{y}^+ = y + h \quad\text{and}\quad y^+ = y - H^{-1}\widehat{\nabla}_y.$$

**Bounding the Newton decrement at $\tilde{y}^+$.** First, we bound the Newton decrement at $\tilde{y}^+$. By definition, the square of the Newton decrement at $\tilde{y}^+ = y + h$ is

$$\lambda(\tilde{y}^+, \Phi) = \nabla\Phi(y + h)^\top\nabla^{-2}\Phi(y + h)\nabla\Phi(y + h).$$

Now for the vector $z$ defined below, we define the function $F$ as

$$z \triangleq \nabla^{-2}\Phi(y + h)\nabla\Phi(y + h) \quad\text{and}\quad F(y) := \nabla\Phi(y)^\top z. \tag{40}$$

The partial derivative of $F$ in direction $h$ is given by

$$\begin{aligned}
DF(y)[h] &= -h^\top\nabla^2\Phi(y)z, \\
&= -\nabla\Phi(y)^\top H^{-1}\nabla^2\Phi(y)z, \\
&= -\nabla\Phi(y)^\top H^{-1/2}H^{-1/2}\nabla^2\Phi(y)H^{-1/2}H^{1/2}z, \\
&= -\nabla\Phi(y)^\top H^{-1/2}(H^{-1/2}\nabla^2\Phi(y)H^{-1/2} - I)H^{1/2}z - \nabla\Phi(y)^\top z. \tag{41}
\end{aligned}$$

Now, by (21), we have

$$\|H^{-1/2}\nabla^2\Phi(y)H^{-1/2} - I\| \le \frac{\alpha}{1 - \alpha}.$$

Thus, the first term on the right-hand side of (41) can be bounded as

$$\nabla\Phi(y)^\top H^{-1/2}(H^{-1/2}\nabla^2\Phi(y)H^{-1/2} - I)H^{1/2}z$$

$$\leq \frac{\alpha}{1-\alpha}\|\nabla\Phi(y)^\top H^{-1/2}\|\|H^{1/2}z\|,$$

$$= \frac{\alpha}{1-\alpha}\|\nabla\Phi(y)\|_{H^{-1}}\|z\|_H,$$

$$\leq \frac{\alpha(1+\alpha)}{(1-\alpha)^2}\|\nabla\Phi(y)\|_{\nabla^{-2}\Phi(y)}\|z\|_{\nabla^2\Phi(y)},$$

$$= \frac{\alpha(1+\alpha)}{(1-\alpha)^2}\lambda(y,\Phi)\cdot\|z\|_{\nabla^2\Phi(y)}.$$

Plugging this into (41) and using the definition $z$ in (40), we obtain

$$\left|DF(y)[h] + F(y)\right| \leq \frac{\alpha(1+\alpha)}{(1-\alpha)^2}\lambda(y,\Phi)\cdot\|z\|_{\nabla^2\Phi(y)}. \tag{42}$$

Now, let $\boldsymbol{y}(s) \coloneqq y + sh$ and $F\circ\boldsymbol{y}(s) \coloneqq F(y(s))$. With this, we have

$$(F\circ\boldsymbol{y})'(s) - (F\circ\boldsymbol{y})'(0) = h^\top(\nabla^2\Phi(y(s)) - \nabla^2\Phi(y(0)))z. \tag{43}$$

On the other hand, by Lemma 8 and our assumption on $\lambda(y,\Phi)$, we have

$$\|\boldsymbol{y}(s) - y\|_{\nabla^2\Phi(y)} = s\|h\|_{\nabla^2\Phi(y)} \leq \frac{s}{1-\alpha}\|\nabla\Phi(y)\|_{\nabla^{-2}\Phi(y)} \leq \frac{1}{1-\alpha}\lambda(y,\Phi), \tag{44}$$

$$< \frac{1}{30M_\Phi}. \tag{45}$$

Thus, by Lemma 1, we have

$$(1 - M_\Phi\|\boldsymbol{y}(s) - y\|_{\nabla^2\Phi(y)}^2)^2\nabla^2\Phi(y) \preccurlyeq \nabla^2\Phi(y(s)) \preccurlyeq \frac{1}{(1 - M_\Phi\|\boldsymbol{y}(s) - y\|_{\nabla^2\Phi(y)})^2}\nabla^2\Phi(y).$$

This, together with (45) also implies that

$$(1 - 3M_\Phi\|\boldsymbol{y}(s) - y\|_{\nabla^2\Phi(y)})\nabla^2\Phi(y) \preccurlyeq \nabla^2\Phi(y(s)) \preccurlyeq (1 + 3M_\Phi\|\boldsymbol{y}(s) - y\|_{\nabla^2\Phi(y)})\nabla^2\Phi(y).$$

After rearranging, this becomes

$$-3M_\Phi\|\boldsymbol{y}(s) - y\|_{\nabla^2\Phi(y)}\nabla^2\Phi(y) \preccurlyeq \nabla^2\Phi(y(s)) - \nabla^2\Phi(y) \preccurlyeq 3M_\Phi\|\boldsymbol{y}(s) - y\|_{\nabla^2\Phi(y)}\nabla^2\Phi(y).$$

Combining this with (44) and the fact that $c \leq \frac{1}{4}$ gives

$$-4M_\Phi\lambda(y,\Phi)\nabla^2\Phi(y) \preccurlyeq \nabla^2\Phi(y(s)) - \nabla^2\Phi(y) \preccurlyeq 4M_\Phi\lambda(y,\Phi)\nabla^2\Phi(y). \tag{46}$$

Finally, by Lemma 9 and (46), we obtain the following bound on the right-hand side of (43):

$$(F\circ\boldsymbol{y})'(s) - (F\circ\boldsymbol{y})'(0) \leq 6M_\Phi\lambda(y,\Phi)\|h\|_{\nabla^2\Phi(y)}\|z\|_{\nabla^2\Phi(y)}.$$

Integrating this over $s$ gives

$$\nabla\Phi(y + h)^\top z = (F\circ\boldsymbol{y})(1)$$

$$= (F\circ\boldsymbol{y})(0) + (F\circ\boldsymbol{y})'(0) + \int_0^1\left((F\circ\boldsymbol{y})'(s) - (F\circ\boldsymbol{y})'(0)\right)ds$$

$$\leq \nabla\Phi(y)^\top z + DF(y)[z] + 6M_\Phi\lambda(y,\Phi)\|h\|_{\nabla^2\Phi(y)}\|z\|_{\nabla^2\Phi(y)},$$

$$\leq \nabla\Phi(y)^\top z + DF(y)[z] + \frac{6M_\Phi}{1-\alpha}\lambda(y,\Phi)^2\|z\|_{\nabla^2\Phi(y)}, \tag{47}$$

where the last inequality follows by (44). Now, note that from (46) (with $s = 1$) and the assumption that $\lambda(w,\Phi) \leq 1/(40M_\Phi)$, we have

$$\nabla^2\Phi(y) \preccurlyeq \frac{10}{9}\nabla^2\Phi(y + h).$$

This implies

$$\|z\|_{\nabla^2\Phi(y)} \le \frac{10}{9}\|z\|_{\nabla^2\Phi(y+h)} = \frac{10}{9}\lambda(y+h,\Phi).$$

Plugging this into (47) and using (42), we get

$$\nabla\Phi(y+h)^\top z \le \left|\nabla\Phi(y)^\top z + DF(y)[h]\right| + \frac{20M_\Phi}{3(1-\alpha)}\lambda(y,\Phi)^2\lambda(y+h,\Phi)$$

$$\le \frac{\alpha(1+\alpha)}{(1-\alpha)^2}\lambda(y,\Phi)\|z\|_{\nabla^2\Phi(y)} + \frac{20M_\Phi}{3(1-\alpha)}\lambda(y,\Phi)^2\lambda(y+h,\Phi),$$

$$\le \frac{10\alpha(1+\alpha)}{9(1-\alpha)^2}\lambda(y,\Phi)\lambda(y+h,\Phi) + \frac{20M_\Phi}{3(1-\alpha)}\lambda(y,\Phi)^2\lambda(y+h,\Phi).$$

Now, from the definition of $z$, we have

$$\nabla\Phi(y+h)^\top z = \lambda(y+h,\Phi)^2.$$

Thus, since $\alpha < 1/4$, we finally get

$$\lambda(y+h,\Phi) \le 9M_\Phi\lambda(y,\Phi)^2 + 2.5\alpha\lambda(y,\Phi).$$

This proves the first part of the claim, i.e. (43).

**Bounding the Newton decrement at $y^+$.** We now bound the Newton decrement at $y^+ = y - H^{-1}\widehat{\nabla}_y$ in terms of that of $\tilde{y}^+ = y + h$. Note that

$$\tilde{y}^+ - y^+ = H^{-1}(\nabla\Phi(y) - \widehat{\nabla}_y).$$

On the other hand, from (64) and (45), we have

$$\nabla^2\Phi(\tilde{y}^+) \preccurlyeq (1 + 3M_\Phi\|h\|)\nabla^2\Phi(y) \preccurlyeq \frac{7}{4}\nabla^2\Phi(y) \preccurlyeq \frac{7}{4}(1+\alpha)H, \tag{48}$$

which implies that

$$\|\nabla^2\Phi(\tilde{y}^+)^{1/2}H^{-1}\nabla^2\Phi(\tilde{y}^+)^{1/2}\| \le \frac{7}{4}(1+\alpha).$$

Therefore,

$$\|\tilde{y}^+ - y^+\|^2_{\nabla^2\Phi(\tilde{y}^+)}$$
$$= (\nabla\Phi(y) - \widehat{\nabla}_y)^\top H^{-1}\nabla^2\Phi(\tilde{y}^+)H^{-1}(\nabla\Phi(y) - \widehat{\nabla}_y),$$
$$= (\nabla\Phi(\tilde{y}^+) - \widehat{\nabla}_y)^\top\nabla^{-1/2}\Phi(\tilde{y}^+)\left(\nabla^2\Phi(\tilde{y}^+)^{1/2}H^{-1}\nabla^2\Phi(\tilde{y}^+)^{1/2}\right)^2\nabla^{-1/2}\Phi(\tilde{y}^+)(\nabla\Phi(y) - \widehat{\nabla}_y),$$
$$\le \frac{49}{16}(1+\alpha)^2(\nabla\Phi(y) - \widehat{\nabla}_y)^\top\nabla^{-2}\Phi(\tilde{y}^+)(\nabla\Phi(y) - \widehat{\nabla}_y),$$
$$= \frac{49}{16}(1+\alpha)^2\|\nabla\Phi(y) - \widehat{\nabla}_y\|_{\nabla^{-2}\Phi(\tilde{y}^+)}.$$

Combining this with (48) and our assumption on $\widehat{\nabla}_y$ from (20) implies

$$\|\tilde{y}^+ - y^+\|_{\nabla^2\Phi(\tilde{y}^+)} \le \frac{7}{2}(1+\alpha)\|\nabla\Phi(y) - \widehat{\nabla}_y\|_{\nabla^{-2}\Phi(y)} \le 5(1+\alpha)\varepsilon. \tag{49}$$

Thus, by Lemma 1, we have

$$\left((1 - 5(1+\alpha)\varepsilon M_\Phi)^2 - 1\right)\nabla^2\Phi(y^+) \preccurlyeq \nabla^2\Phi(\tilde{y}^+) - \nabla^2\Phi(y^+) \le \left(\frac{1}{(1 - 5(1+\alpha)\varepsilon M_\Phi)^2} - 1\right)\nabla^2\Phi(y^+).$$

Since $\varepsilon < 1/(40M_\Phi)$, we get

$$-20(1+\alpha)\varepsilon\nabla^2\Phi(y^+) \preccurlyeq \nabla^2\Phi(\tilde{y}^+) - \nabla^2\Phi(y^+) \preccurlyeq 20(1+\alpha)\varepsilon\nabla^2\Phi(y^+). \tag{50}$$

Now, by Lemma 4 instantiated with $x = \tilde{y}^+$ and $w = y^+$, we have

$$\sqrt{(\nabla\Phi(\tilde{y}^+) - \nabla\Phi(y^+))^\top \nabla^{-2}\Phi(\tilde{y}^+)(\nabla\Phi(\tilde{y}^+) - \nabla\Phi(y^+))} \leq \frac{\|y^+ - \tilde{y}^+\|_{\nabla^2\Phi(\tilde{y}^+)}}{(1 - M_\Phi\|y^+ - \tilde{y}^+\|_{\nabla^2\Phi(\tilde{y}^+)})},$$

$$\leq 10(1 + \alpha)\varepsilon, \tag{51}$$

where in the last inequality we used (49) and the fact that $\varepsilon \leq 1/(40M_\Phi)$.

Using the triangle inequality, we can bound the Newton decrement at $y^+$ as

$$\lambda(y^+, \Phi) \leq \sqrt{(\nabla\Phi(\tilde{y}^+) - \nabla\Phi(y^+))^\top \nabla^{-2}\Phi(y^+)(\nabla\Phi(\tilde{y}^+) - \nabla\Phi(y^+))}$$
$$+ \sqrt{\nabla\Phi(\tilde{y}^+)^\top \nabla^{-2}\Phi(y^+)\nabla\Phi(\tilde{y}^+)},$$
$$\leq 2\sqrt{(\nabla\Phi(\tilde{y}^+) - \nabla\Phi(y^+))^\top \nabla^{-2}\Phi(\tilde{y}^+)(\nabla\Phi(\tilde{y}^+) - \nabla\Phi(y^+))}$$
$$+ \sqrt{\nabla\Phi(\tilde{y}^+)^\top \nabla^{-2}\Phi(y^+)\nabla\Phi(\tilde{y}^+)}, \quad \text{(by (50) and } \varepsilon \leq 1/(40M_\Phi))$$
$$\leq 20(1 + \alpha)\varepsilon + (1 + 20(1 + \alpha)\varepsilon) \cdot \lambda(\tilde{y}^+, \Phi),$$

where the last inequality follows by (50) and (51). This completes the proof. $\qquad\square$

## D.2 Proof of Lemma 6

**Proof.** By definition of $(w_t^m)$ in Algorithm 1, we have $w_t^1 = w_{t-1}$ and $w_t = w_t^{m_{\text{Newton}}}$. We show properties (23), (25), (26), and 27 using induction over $m = 1, \ldots, m_{\text{Newton}}$.

**Base case.** We start with the base case; $m = 1$. Note that from the assumption in (22) and definition of $w_t^1$, we have

$$\|w_t^1 - u_{t-1}\|_{\nabla^2\Phi(u_{t-1})} \leq 1/(40M_\phi). \tag{52}$$

Now, by definition of the Oracle $\mathcal{O}_\alpha^{\text{hess}}$ and the fact that $H_{t-1} = \mathcal{O}_\alpha^{\text{hess}}(u_{t-1})$ (see Algorithm 1) with $\alpha = 0.001$, we have

$$(1 - 0.001)\nabla^2\Phi(u_{t-1}) \preccurlyeq H_{t-1} \preccurlyeq (1 + 0.001)\nabla^2\Phi(u_{t-1}). \tag{53}$$

Combining this with (52) implies property (25) for the base case. Furthermore, since $w_t^1 = w_{t-1}$ (by definition), (27) follows trivially for the base case.

Now, using that $\Phi_t(w) = \Phi_{t-1}(w) + \eta g_{t-1}^\top w$, we have

$$\lambda(w_t^1, \Phi_t)^2 = \lambda(w_{t-1}, \Phi_t)^2$$
$$= (\nabla\Phi_{t-1}(w_{t-1}) + \eta g_{t-1})^\top \nabla^{-2}\Phi(w_{t-1})(\nabla\Phi_{t-1}(w_{t-1}) + \eta g_{t-1})$$
$$\leq 2\nabla\Phi_{t-1}(w_{t-1})^\top \nabla^{-2}\Phi(w_{t-1})\nabla\Phi_{t-1}(w_{t-1}) + 2\eta^2 g_{t-1}^\top \nabla^{-2}\Phi(w_{t-1})g_{t-1}$$
$$= 2\lambda(w_{t-1}, \Phi_{t-1})^2 + 2\eta^2 g_{t-1}^\top \nabla^{-2}\Phi(w_{t-1})g_{t-1} \tag{54}$$
$$\leq 2\alpha^2 + 2\eta^2 b^2 \leq 1/(2500M_\Phi^2), \tag{55}$$

where the last inequality follows by (22) and the fact that $\|g_{t-1}\|_{\nabla^{-2}\Phi(w_{t-1})} \leq b$. This shows property (23) for the base case. Thus, by Lemma 1, we have, for $w_t^\star \in \operatorname{argmin}_{w \in \mathcal{K}} \Phi_t(w)$,

$$\|w_t^1 - w_t^\star\|_{\nabla^2\Phi(w_t^1)} = \|w_t^1 - w_t^\star\|_{\nabla^2\Phi_t(w_t^1)} \leq \lambda(w_t^1, \Phi_t)/(1 - M_\Phi\lambda(w_t^1, \Phi_t)) \leq \frac{1}{49M_\Phi}. \tag{56}$$

Now, combining (53) with the fact that $\|u_{t-1} - w_t^1\|_{\nabla^2\Phi(u_{t-1})} = \|u_{t-1} - w_{t-1}\|_{\nabla^2\Phi(u_{t-1})} \leq 1/(40M_\Phi)$ (see (22)) and Lemma 2, we obtain

$$\left(\frac{31}{32}\right)^2 H_{t-1} \preccurlyeq \nabla^2\Phi(w_t^1) \preccurlyeq \left(\frac{32}{31}\right)^2 H_{t-1}, \tag{57}$$

Plugging (57) into (56), we get

$$\|w_t^1 - w_t^\star\|_{H_{t-1}} \leq 1/(40M_\Phi). \tag{58}$$

Now, by the triangle inequality

$$\|u_{t-1} - w_t^\star\|_{H_{t-1}} \le \|u_{t-1} - w_t^1\|_{H_{t-1}} + \|w_t^1 - w_t^\star\|_{H_{t-1}}$$
$$\le 1/(20\Phi_M), \tag{59}$$

where in the last inequality we used (22) and (58). Combining (59) with (53) and Lemma 1, we get

$$(4/5)\nabla^2\Phi(w_t^\star) \preccurlyeq H_{t-1} \preccurlyeq (5/4)\nabla^2\Phi(w_t^\star). \tag{60}$$

Combining Equations (60) and (57) implies property (24) for the base of Induction. Furthermore, note that from Lemma 1:

$$\|w_t^1 - w_t^\star\|_{\nabla^2\Phi(w_t^\star)} \le \frac{50}{49}\lambda(w_t^1, \Phi_t) \le \frac{1}{49M_\Phi},$$

which shows property (26) for the base case.

**Induction step.** Now, assume that properties (23), (25), (26), and (27) hold for $m \ge 1$. We will show that these properties holds for $m+1$. From the hypothesis of induction, we have

$$\|w_t^m - u_{t-1}\|_{H_{t-1}} \le \frac{1}{12M_\Phi},$$

which combined with (53) and Lemma 1 implies

$$0.84\nabla^2\Phi(w_t^m) \preccurlyeq H_{t-1} \preccurlyeq 1.2\nabla^2\Phi(w_t^m). \tag{61}$$

Thus, by Lemma 5 (instantiated with $c = 1/5$) and the fact that $\lambda(w_t^m, \Phi_t) \le 1/(40M_\Phi)$ (by the induction hypothesis), we get that for $\tilde{w}_t^{m+1} \coloneqq w_t^m - H_{t-1}^{-1}\nabla\Phi_t(w_t^m)$:

$$\lambda(\tilde{w}_t^{m+1}, \Phi_t) \le 9M_\Phi\lambda(w_t^m, \Phi_t)^2 + 2.5c\lambda(w_t^m, \Phi_t) \le (7/8)\lambda(w_t^m, \Phi_t).$$

Again, by Lemma 5 with $c = 1/5$, we have

$$\lambda(w_t^{m+1}, \Phi_t) \le 20(1+c)\varepsilon + (1 + 20(1+c)\varepsilon)\lambda(\tilde{w}_t^{m+1}, \Phi)$$
$$\le 25\varepsilon + \left(\frac{15}{16}\right)\lambda(w_t^m, \Phi_t) \le 1/(50M_\Phi). \tag{62}$$

By the induction hypothesis, we also have that $\lambda(w_t^m, \Phi_t) \le \left(\frac{15}{16}\right)^{m-1}\lambda(w_t^1, \Phi_t) + 500\varepsilon$. Combining this with (62), we get

$$\lambda(w_t^{m+1}, \Phi_t) \le \left(\frac{15}{16}\right)^m\lambda(w_t^1, \Phi_t) + 500\varepsilon. \tag{63}$$

This shows that (23) holds with $m$ replaced by $m+1$.

Next, we show that (25) holds with $m$ replaced by $m+1$. Combining (62) with Lemma 1 implies

$$\|w_t^{m+1} - w_t^\star\|_{\nabla^2\Phi(w_t^\star)} \le \lambda(w_t^{m+1}, \Phi_t)/(1 - M_\Phi\lambda(w_t^{m+1}, \Phi_t)) \le 1/(49M_\Phi).$$

This, together with (60) gives

$$\|w_t^{m+1} - w_t^\star\|_{H_{t-1}} \le 1/(32M_\Phi). \tag{64}$$

Combining (59) with (64) gives:

$$\|w_t^{m+1} - u_{t-1}\|_{H_{t-1}} \le \frac{1}{12M_\Phi},$$

which proves that (25) holds with $m$ replaced by $m+1$.

Next, we show that (26) holds with $m$ replaced by $m+1$. By (63) and Lemma 1, we have

$$\|w_t^{m+1} - w_t^\star\|_{\nabla^2\Phi(w_t^\star)} \le \frac{50}{49}\left(\frac{15}{16}\right)^m\lambda(w_t^1, \Phi_t) + 240\varepsilon,$$
$$\le \frac{1}{49M_\Phi}\left(\frac{15}{16}\right)^m + 240\varepsilon \le \frac{1}{20M_\Phi}, \tag{65}$$

where in the last inequality we used (55) and the bound on $\varepsilon$ in the lemma's statement. This shows that (26) holds with $m$ replaced by $m + 1$.

Next, we show that (27) holds with $m$ replaced by $m + 1$. By plugging (60) into (65), we get

$$\|w_t^{m+1} - w_t^\star\|_{H_{t-1}} \leq \frac{1}{40M_\Phi}\left(\frac{15}{16}\right)^m + 300\varepsilon. \tag{66}$$

On the other hand, by (54), we have

$$\lambda(w_t^1, \Phi_t) \leq \sqrt{\nabla\Phi_{t-1}(w_{t-1})^\top(\nabla^2\Phi(w_{t-1}))^{-1}\nabla\Phi_{t-1}(w_{t-1})} + \sqrt{\eta^2 g_{t-1}^\top(\nabla^2\Phi(w_{t-1}))^{-1}g_{t-1}}$$
$$= \lambda(w_{t-1}, \Phi_{t-1}) + \eta\|g_{t-1}\|_{\nabla^{-2}\Phi(w_{t-1})}.$$

Plugging this into (56) and using (55), we get

$$\|w_t^1 - w_t^\star\|_{\nabla^2\Phi(w_t^1)} \leq \frac{50}{49}(\lambda(w_{t-1}, \Phi_{t-1}) + \eta\|g_{t-1}\|_{\nabla^{-2}\Phi(w_{t-1})}),$$
$$\leq \frac{50}{49}(\alpha + \eta\|g_{t-1}\|_{\nabla^{-2}\Phi(w_{t-1})}), \tag{67}$$

where the last inequality follows by (22). Combining (67) with (61) (instantiated with $m = 1$),

$$\|w_t^1 - w_t^\star\|_{H_{t-1}} \leq 2(\alpha + \eta\|g_{t-1}\|_{\nabla^{-2}\Phi(w_{t-1})}). \tag{68}$$

Now, by (66), (68), and the triangle inequality, we have

$$\|w_t^1 - w_t^{m+1}\|_{H_{t-1}} \leq 2\eta\|g_{t-1}\|_{\nabla^{-2}\Phi(w_{t-1})} + \frac{1}{40M_\Phi}\left(\frac{15}{16}\right)^m + 500\varepsilon + 2\alpha.$$

Via another triangle inequality, we get

$$\left|\|u_{t-1} - w_t^{m+1}\|_{H_{t-1}} - \|u_{t-1} - w_{t-1}\|_{H_{t-1}}\right| \leq 2\eta\|g_{t-1}\|_{\nabla^{-2}\Phi(w_{t-1})} + \frac{1}{40M_\Phi}\left(\frac{15}{16}\right)^m$$
$$+ 500\varepsilon + 2\alpha.$$

This shows that (27) holds with $m$ replaced by $m + 1$. Finally, combining (60) with (61) implies

$$\frac{1}{2}\nabla^2\Phi(w_t^m) \preccurlyeq \nabla^2\Phi(w_t^\star) \preccurlyeq 2\nabla^2\Phi(w_t^m),$$

which completes the proof. $\qquad\square$

# E  Helper Lemmas

**Lemma 8** (Bounding norm of the Newton step). *Let $y \in \mathcal{K}$ and $H \in \mathbb{R}^{d \times d}$ be such that $(1 - c)\nabla^2\Phi(y) \preccurlyeq H \preccurlyeq (1 + c)\nabla^2\Phi(y)$. Then, for $h := -H^{-1}\nabla\Phi(y)$, we have*

$$\|h\|_{\nabla^2\Phi(y)} \leq \frac{1}{1-c}\|\nabla\Phi(y)\|_{\nabla^{-2}\Phi(y)}.$$

**Proof.** We can write

$$\|h\|_{\nabla^2\Phi(y)} = \|H^{-1}\nabla\Phi(y)\|_{\nabla^2\Phi(y)}$$
$$= \sqrt{\nabla\Phi(y)^\top H^{-1}\nabla^2\Phi(y)H^{-1}\nabla\Phi(y)}$$
$$= \sqrt{\nabla\Phi(y)^\top\nabla^2\Phi(y)^{-1/2}\left(\nabla^2\Phi(y)^{1/2}H^{-1}\nabla^2\Phi(y)^{1/2}\right)^2\nabla^2\Phi(y)^{-1/2}\nabla\Phi(y)}. \tag{69}$$

For the middle matrix $\nabla^2\Phi(y)^{1/2}H^{-1}\nabla^2\Phi(y)^{1/2}$ we have that

$$\frac{1}{1+c}I \preccurlyeq \nabla^2\Phi(y)^{1/2}H^{-1}\nabla^2\Phi(y)^{1/2} \preccurlyeq \frac{1}{1-c}I,$$

since $(1 - c)\nabla^2\Phi(y) \preccurlyeq H \preccurlyeq (1 + c)\nabla^2\Phi(y)$ by assumption. Plugging this back into (69), we get

$$\|h\|_{\nabla^2\Phi(y)} \leq \frac{1}{1-c}\|\nabla\Phi(y)\|_{\nabla^2\Phi(y)^{-1}}.$$

$\qquad\square$

**Lemma 9** (Cauchy-Schwarz). *If $-B \preccurlyeq A \preccurlyeq B$ are symmetric matrices and $B$ is PSD, then*
$$x^\top A y \leq \|x\|_B \|y\|_B.$$

