# OpenReview forum: "Projection-Free Online Convex Optimization via Efficient Newton Iterations"
_NeurIPS.cc/2023/Conference — NeurIPS 2023 poster_

### Official Review · Reviewer_y1W5 · 2023-06-09

**Soundness:** 3 good
**Presentation:** 2 fair
**Contribution:** 3 good
**Rating:** 7
**Confidence:** 4

**Summary:**

Building upon recent progress in projection-free OCO, this paper presents a new IPM-like algorithm attaining optimal regret with improved efficiency, which calculates Hessians only $O(\sqrt{T})$ number of times. The algorithm assumes access to a self-concordant function by gradient oracles and Hessian oracles.

**Strengths:**

This paper makes a solid contribution on more efficient projection-free OCO, which admits optimal regret and better efficiency at the same time. Moreover, it also opens a new avenue of using IPM-like methods for projection-free OCO, besides classic LOO-based methods and SO/MO-based approaches developed recently.

**Weaknesses:**

There is an author's comment in line 160, which I suspect violates the double-blind rule. Though the contents are interesting, I regret to reject the paper for this reason.

I think the paper can benefit from a detailed comparison with the ONS algorithm in HAK07.

There are many typos, here are some that I caught:

Line 12: expansive - expensive (it appears multiple places)

Line 39: insure - ensure

Line 129: interior - interior of

Line 13 in the alg box: there should be another line of Set $H_{t+1}=H_t$?

---Update---

After a discussion with the AC, the comment is considered as a typo and I changed the score accordingly.

**Questions:**

Can you elaborate more on how to obtain Corollary 1?

**Limitations:**

There is an author's comment in line 160, which I suspect violates the double-blind rule.

---

> ### Author Rebuttal · Authors · 2023-08-10
>
> Thank you for your positive review.
>
> **“There is an author's comment in line 160, which I suspect violates the double-blind rule...”**
> Unfortunately, this was a typo that we only spotted shortly after the submission deadline.
>
> **“I think the paper can benefit from a detailed comparison with the ONS algorithm in HAK07.”**
> We note that the ONS algorithm in [Hazan et al. 2007] is an algorithm for online exp-concave optimization and is not intended for use in the OCO setting we consider in this paper; naively applying ONS to OCO would lead to suboptimal (even linear) regret. This is because ONS  uses certain quadratic terms (designed for the exp-concave setting) in the regularizer that can lead to a linear regret in the OCO setting. Furthermore, the ONS algorithm requires generalized projections at each iteration which is precisely what we are trying to avoid as these can be computationally costly; see discussion in [Garber and Kretzu 2023].
>
> **“There are many typos, here are some that I caught”**
> The typos will be fixed in the revision. And you are correct, there should be $H_{t+1}=H_t$ on line 13 of the algo (we have already spotted this and fixed it).
>
>
> **“Can you elaborate more on how to obtain Corollary 1?”**
> When using the LS barrier, results in [Lee and Sidford 2019] imply that we have
> - $\mathcal{C}^{\texttt{grad}}_{\varepsilon}\leq \widetilde{O}(\mathcal{C}^{\texttt{sys}} \cdot \log (\varepsilon^{-1}))$; and
> - $\mathcal{C}^{\texttt{hess}}_{\varepsilon} = \widetilde{O}(\mathcal{C}^{\texttt{sys}} \sqrt{d} \cdot  \log (\varepsilon^{-1}))$,
>
> where $\mathcal{C}^{{\texttt{sys}}}$ is the computational cost of solving a linear system of the form $A^\top \text{diag}(v) A x = y$, for vectors $v\in \mathbb{R}^{d}_{\geq 0}$ and $y\in \mathbb{R}^d$; here $A$ represents the constraint matrix of the polytope. In the worst-case, the linear-system solve cost is $O(m d^{\omega -1})$ (i.e. $\mathcal{C}^{{\texttt{sys}}}\leq O(m d^{\omega -1})$), where $\omega$ is the exponent of matrix multiplication. Plugging these bounds into the computational complexity in Theorem 6 should imply the claimed complexity in Corollary 1. We will add these details in the revision.
>
> **References:**
>
> Elad Hazan, Amit Agarwal, and Satyen Kale. "Logarithmic regret algorithms for online convex optimization." Machine Learning 69 (2007): 169-192.
>
> Yin Tat Lee, and Aaron Sidford. "Solving linear programs with sqrt (rank) linear system solves." arXiv preprint arXiv:1910.08033 (2019).
>
> Dan Garber and Ben Kretzu. "Projection-free Online Exp-concave Optimization." arXiv preprint arXiv:2302.04859 (2023).

---

> > ### Comment · Reviewer_y1W5 · 2023-08-11
> > **Reply to Authors**
> >
> > Thank you for your reply.
> >
> > By comparison with ONS I meant what you wrote here, and I believe such discussion can make the contributions of this work more clear.
> >
> > As for Corollary 1, I still don't get where the $d^{\frac{7}{2}}$ terms comes from. Plugging these bounds into Theorem 6, the computation seems to be $\tilde{O}(md^{w-1}T+d^2T+md^w\sqrt{T})$.

---

> > > ### Author Response · Authors · 2023-08-12
> > > **Clarifying the complexity in Corollary 1**
> > >
> > > That is indeed the correct complexity, which may further be simplified to $\widetilde{O}(m d^{\omega -1} T + m d^{\omega} \sqrt{T})$, if we assume that $d^2 \leq m d^{\omega-1}$; $m\geq d$ is the interesting setting since otherwise one can just use the standard log-barrier instead of the LS-barrier. Corollary 1 displays $d^{7/2} \sqrt{T}$ instead of $m d^{\omega} \sqrt{T}$, for the lower-order term. I believe this was just a typo.

---

> > > > ### Comment · Reviewer_y1W5 · 2023-08-13
> > > >
> > > > Thanks for the confirmation. Please add these details to Corollary 1.

---

> > > > > ### Author Response · Authors · 2023-08-13
> > > > > **Acknowledgment**
> > > > >
> > > > > Thanks. We will make sure to add these details.

---

### Official Review · Reviewer_eL2Y · 2023-06-17

**Soundness:** 3 good
**Presentation:** 3 good
**Contribution:** 2 fair
**Rating:** 6
**Confidence:** 3

**Summary:**

This paper introduces new projection-free algorithms for online convex optimization, which utilize Newton iterates with a self-concordant barrier for the target set. The authors establish a state-of-the-art regret bound for this algorithm.

**Strengths:**

Strengths And Weaknesses:
The paper presents a new approach for projection-free online convex optimization. The main advantage of the proposed algorithm is that it only requires computing a full inverse of the Hessian in a vanishing O(1/\sqrt{T}) fraction of each round. Additionally, for the case of a polytope with m constraints, their method exhibits lower per-iteration computational cost compared to linear optimization. Furthermore, their method achieves a better regret bound than existing works in this specific scenario.

However, I have some concerns that I would like the authors to address:

It would be beneficial to have a comparison of the gradient complexity between this method and other related works. Additionally, it would be valuable to provide further analysis of the computational complexity mentioned in line 208. Specifically, it would be helpful to clarify under which circumstances the gradient complexity dominates the Hessian complexity, and vice versa.

Regarding the computation complexity mentioned in line 208, there is a coefficient M_{\Phi}. In some cases, M_{\Phi} can be significantly larger than d. For instance, when considering a polytope with m constraints, M_{\Phi} is related to m, and m can be exponential in d.

In Theorem 6, the authors assume that the local norm of g_t is less than C for all w_t. This assumption is not trivial, as the Hessian of a self-concordant function is typically unbounded.

**Weaknesses:**

See above.

**Questions:**

See above.

**Limitations:**

Not applicable.

---

> ### Author Rebuttal · Authors · 2023-08-10
>
> Thank you for your review.
>
> **“It would be beneficial to have a comparison of the gradient complexity between this method and other related works.”**
> In previous works, the gradient complexity refers to the number of gradient computations of the losses. Our algorithm only requires a single gradient computation of the loss at each step, which is on par with previous approaches. We suspect that you are referring to $\mathcal{C}^{\texttt{grad}}$. We answer this next.
>
> **“Additionally, it would be valuable to provide further analysis of the computational complexity mentioned in line 208. Specifically, it would be helpful to clarify under which circumstances the gradient complexity dominates the Hessian complexity, and vice versa.…”**
> The Hessian cost $\mathcal{C}^{\texttt{hess}}$ will typically dominate the gradient cost $\mathcal{C}^{\texttt{grad}}$ for the barriers of essentially all common convex sets used in practice. To get a sense of this, let’s look at the case of a polytope in $\mathbb{R}^d$ with $m$ constraints (these details will be added in the revision). When using a standard log-barrier $\Phi$, we have $M_{\Phi}=1$, $\mathcal{C}^{\texttt{grad}}\leq O(md)$, and $\mathcal{C}^{\texttt{hess}} = O(m d^{\omega -1})$ (these are the costs of computing the gradients and Hessians exactly), where $\omega$ is the exponent of matrix multiplication. Now, if we use the LS barrier as in Section 4 (which confers benefits in terms of the regret), results in [Lee and Sidford 2019] imply that we have
> - $M_{\Phi}=O(\log(m)^{2/5})$;
> - $\mathcal{C}^{\texttt{grad}}_{\varepsilon}\leq \widetilde{O}(\mathcal{C}^{\texttt{sys}} \cdot \log(\varepsilon^{-1}))$; and
> - $\mathcal{C}^{\texttt{hess}}_{\varepsilon} = \widetilde{O}(\mathcal{C}^{\texttt{sys}} \sqrt{d} \cdot \log(\varepsilon^{-1}))$,
>
>  where $\mathcal{C}^{{\texttt{sys}}}$ is the computational cost of solving a linear system of the form $A^\top \text{diag}(v) A x = y$, for vectors $v\in \mathbb{R}^{d}_{\geq 0}$ and $y\in \mathbb{R}^d$; here $A$ represents the constraint matrix of the polytope. So for both the LS- and log-barrier, the cost of approximate Hessian evalution dominates that of (approximate) gradient evaluation. We will add details on this in the revision.
>
> **“Regarding the computation complexity mentioned in line 208, there is a coefficient M_{\Phi}...”**
> As we mentioned in the previous paragraph (see also display after Line 229), the constant $M_\Phi$ for the LS barrier [resp. Log-barrier] is such that $M_{\Phi}=O(\log(m)^{2/5})$ [resp. $M_{\Phi}=1$]. And so for the application to polytopes, $M_{\Phi}$ is at most logarithmic in $m$ (and so at most polynomial in $d$ when $m$ is exponential in $d$). Note that $M_{\Phi}$ is not to be confused with the parameter $\nu$ of the self-concordant barrier (see Definition 2), which is the one that may scale polynomially in $m$ depending on the choice of the barrier. In fact, for the log-barrier, we have $\nu =m$, and for the LS barrier we have $\nu =d$; having $\nu$ being independent of $m$ is precisely the appeal of using the LS barrier.
>
> **“In Theorem 6, the authors assume that the local norm of g_t is less than C for all w_t. This assumption is not trivial, as the Hessian of a self-concordant function is typically unbounded.”**
>
> Note that the assumption we make involves the inverse of the Hessian, i.e. we assume $\Vert g_t\rVert_{\nabla^{-2}\Phi(w_t)}$ is bounded.
>
> And so large Hessian eigenvalues only make things better here. In general, you should think of this assumption as being much weaker than the standard Lipschitzness assumption in OCO. In fact, there are many settings where the losses are non-lipshitz, i.e. where $\Vert g_t\rVert$ is unbounded, but the local norm $\Vert g_t\rVert_{\nabla^{-2}\Phi(w_t)}$ is. This includes, for example, the setting of linear regression with the log-loss (see e.g. Section 6 and Lemma 7 in [Rakhlin and Sridharan 2015]), and the classical portfolio selection setting (see e.g. [Luo et al. 2018]). There are other settings where the local gradient norms can be much smaller that their standard Euclidean counterparts; see [Abernethy et al. 2012] for a discussion.
>
>
>
> **References:**
>
> Jacob D. Abernethy, Elad Hazan, and Alexander Rakhlin. "Interior-point methods for full-information and bandit online learning." IEEE Transactions on Information Theory 58, no. 7 (2012): 4164-4175.
>
> Alexander Rakhlin, and Karthik Sridharan. "Sequential probability assignment with binary alphabets and large classes of experts." arXiv preprint arXiv:1501.07340 (2015).
>
> Haipeng Luo, Chen-Yu Wei, and Kai Zheng. "Efficient online portfolio with logarithmic regret." Advances in neural information processing systems 31 (2018).
>
> Yin Tat Lee and Aaron Sidford. "Solving linear programs with sqrt (rank) linear system solves." arXiv preprint arXiv:1910.08033 (2019).

---

> > ### Comment · Reviewer_eL2Y · 2023-08-12
> >
> > Thank you for your response. My concerns has been answered therefore I raised my score.

---

### Official Review · Reviewer_kmvY · 2023-07-07

**Soundness:** 3 good
**Presentation:** 3 good
**Contribution:** 2 fair
**Rating:** 4
**Confidence:** 3

**Summary:**

This paper proposes new projection-free algorithms for online convex optimization over a convex domaim $\mathcal{K}$.
Specifically, this paper proposes efficient Newton iterations to obtain projection-free online convex optimization.

**Strengths:**

This paper proposes an efficient online Newton method which is computation efficient.

**Weaknesses:**

This paper seems only using existing method to the online Newton. Though the author claims that unlike the setting of [26], where it is possible to add quadratic terms to the barrier for additional stability, in our setting, we cannot do that wihout sacrificing performance in terms of reget, we could not find any theoretical hardness in this paper to overcome this problem.

**Questions:**

no

---

> ### Author Rebuttal · Authors · 2023-08-10
>
> Thank you for your review.
>
> **“This paper seems only using existing method to the online Newton.”**
> Our algorithm does apply novel techniques to achieve the desired regret guarantee in the general OCO setting. In fact, applying existing techniques such as those in [Mhammedi and Gatmiry 2023] fails for two reasons:
> The first reason, as you already mentioned, is that we cannot add quadratic terms as in [Mhammedi and Gatmiry 2023] since this would lead to a suboptimal regret in the OCO setting. These terms confer stability to the iterates of their algorithm, and they are able to add them without sacrificing performance because they consider the online exp-concave optimization setting. One novelty of our algorithm is in the choice of regularizer, which ensures stability without any quadratic terms.
> Stability aside, the other issue is that their algorithm performs Taylor expansions to approximate Hessian matrices, which are required to compute the Newton iterates. Note that these Taylor expansions are only computationally efficient when the feasible set is a Euclidean ball, and so this does not apply to our OCO setting with general convex sets. To overcome this issue, we replace Taylor expansions by multiple Newton steps per round (see Lines 6-9 of Algorithm 1), and we show (via a non-trivial analysis) that doing this essentially leads to the same guarantee as one would get using Taylor expansions. As far as we know, this approach of taking multiple Newton steps per round in novel.
>
>
> **References:**
>
> Zakaria Mhammedi and Khashayar Gatmiry. "Quasi-newton steps for efficient online exp-concave optimization." In The Thirty Sixth Annual Conference on Learning Theory, pp. 4473-4503. PMLR, 2023.

---

> > ### Comment · Reviewer_kmvY · 2023-08-12
> >
> > I believe that by introducing the self-concordant barrier, the techinique of [Mhammedi and Gatmiry 2023] can be easily adapted to this paper.

---

> > > ### Author Response · Authors · 2023-08-12
> > > **Reply to kmvY**
> > >
> > > We respectfully disagree. As explained in the rebuttal, the Taylor expansion used in [Mhammedi and Gatmiry 2023] cannot be used with sets other than a Euclidean ball while ensuring computational efficiency (this is not a matter of introducing a self-concordant barrier). Using multiple Newton iterates instead of a Taylor expansion is a non-trivial solution to the problem.

---

### Official Review · Reviewer_WCK3 · 2023-07-09

**Soundness:** 3 good
**Presentation:** 3 good
**Contribution:** 3 good
**Rating:** 8
**Confidence:** 4

**Summary:**

This paper proposes a "projection-free" algorithm for online convex optimization. The proposed algorithm adopts a self-concordant barrier of the constraint set as the regularizer, automatically ensuring the feasibility of the actions. The proposed algorithm only requires computing the inverse of an approximate Hessian at some rounds, instead of all rounds, achieving a small overall complexity when, e.g., the constraint set is a polytope.

---

The authors have addressed my questions. I keep the original rating.

**Strengths:**

1. The algorithm is new and the analysis is non-trivial.
2. The algorithm achieves the currently best regret performance for online convex optimization on a polytope.

**Weaknesses:**

1. **Computation of approximate gradient and Hessian.** I wonder how reasonable the error tolerances in the definitions of the approximate gradient and Hessian are, regarding that the Hessian inverse can have arbitrarily small eigenvalues in general. Can the approximate gradient and Hessian easily computed in general?
2. **Explanations about complexity claims.** The claim that the per-round complexity of the algorithm is cheaper than that of the linear optimization oracle and the complexity bound in Corollary 1 need some explanations.
3. **Inconsistent names.** It is claimed that the Lewis-Weights barrier will be used in Section 1 and the title of Section 4, but Ln. 226 says the barrier is called the Lee-Sidford barrier.
4. **Presentation of Algorithm 1.** In Ln. 12--14, $H_{t + 1}$ is not specified.
5. **Typos.**
    - Ln. 160: "[zm:cite]"
    - Ln. 188: an -> and
    - Ln. 189: turning -> tuning
    - Ln. 190: "additional assumptions on the sequence of losses.": incomplete sentence.
    - Ln. 199: if -> of
    - Ln. 232: $\tilde{O} (1)$ is not an appropriate use of the notation.
    - Ln. 299: "in the sense of (3) *and (3)*"

**Questions:**

Please address the weaknesses.

**Limitations:**

This is a theory paper. The assumptions are explicitly stated. Other possible issues I have noticed have been pointed out in the weaknesses block.

---

> ### Author Rebuttal · Authors · 2023-08-10
>
> Thank you for you positive review and helpful suggestions.
>
> **"I wonder how reasonable the error tolerances in the definitions of the approximate gradient and Hessian are…"**
> To get a sense of this, let’s look at the case of a polytope in $\mathbb{R}^d$ with $m$ constraints. Here, we want to use the LS barrier (see Section 4) as it gives a regret bound that is independent of the number of constraints $m$ (this would not be the case with the standard log-barrier). When using the LS barrier, results in [Lee and Sidford 2019] imply that we have
> - $\mathcal{C}^{\texttt{grad}}_{\varepsilon}\leq \widetilde{O}(\mathcal{C}^{\texttt{sys}} \cdot \log (\varepsilon^{-1}))$, and
> - $\mathcal{C}^{\texttt{hess}}_{\varepsilon} = \widetilde{O}(\mathcal{C}^{\texttt{sys}} \sqrt{d} \cdot  \log (\varepsilon^{-1}))$,
>
> where $\mathcal{C}^{\texttt{sys}}$ is the computational cost of solving a linear system of the form $A^\top \text{diag}(v) A x = y$, for vectors $v\in \mathbb{R}^{d}_{\geq 0}$ and $y\in \mathbb{R}^d$; here $A$ represents the constraint matrix of the polytope. So when using the LS barrier, the computational efficiency is directly related to how efficiently the linear system can be solved. In the worst-case, the linear-system solve cost is $O(m d^{\omega -1})$, where $\omega$ is the exponent of matrix multiplication, but in many practical cases, the cost is much smaller.
>
> We note that if a linear optimization oracle (which is typically used by other projection-free algorithms) is implemented via an interior point method, then the state-of-the-art approach would require $\sqrt{d}$ linear-system solves of the type described in the previous paragraph (see [Lee and Sidford 2019]). This means that our algorithm can be a factor $\sqrt{d}$ faster than linear-optimization-based projection-free algorithms (because our algorithm only requires $\widetilde{O}(1)$ such linear-system solves per round). We will add details on this in the revision (together with clarifications for the complexity in Corollary 1).
>
> **“The claim that the per-round complexity of the algorithm is cheaper than that of the linear optimization oracle”**
> Please see previous paragraph (we will add details in the revision).
>
> **“Inconsistent names” + “resentation of Algorithm 1. In Ln. 12--14, $H_{t+1}$ is not specified.”**
> These are typos that we will fix. Thank you for pointing them out.
>
> **References:**
>
> Yin Tat Lee and Aaron Sidford. "Solving linear programs with sqrt (rank) linear system solves." arXiv preprint arXiv:1910.08033 (2019).

---

> > ### Comment · Reviewer_WCK3 · 2023-08-11
> > **Keeping rating**
> >
> > Thanks for the explanation. Please add these details.

---

> > > ### Author Response · Authors · 2023-08-13
> > > **Acknowledgment**
> > >
> > > Thanks. We will make sure to add these details.

---

### Comment · Area_Chair_RTJK · 2023-08-11
**Discussion period**

Dear reviewers and authors,

Thank you very much for your work on this submission and its evaluation. Now that the authors have responded to the reviews, I *strongly encourage* the reviewers to acknowledge the review, to look at other reviews and rebuttals for this submission, and to adjust their scores if needed. Thanks to those that have already done so.

Authors have the possibility to reply if further questions are needed, until the 16th.

Thank you very much to all,

Area Chair

---

### Decision · Program_Chairs · 2023-09-21

**Decision:**

Accept (poster)

**Comment:**

There is an overall consensus among reviewers that this is an interesting submission that should be accepted, which is also my evaluation.